# SEMI-HYPERGRAPH BENCHMARK: ENHANCING FLEXIBILITY OF HYPERGRAPH LEARNING WITH DATASETS AND BENCHMARKS

## ABSTRACT

Graphs are widely used to encapsulate a variety of data formats, but real-world networks often involve complex node relations beyond only being pairwise. While hypergraphs have been developed and employed to account for the complex node relations, they reduce the flexibility of machine learning systems by totally disregarding simple edges, which to some extent leads to a drop in performance. Additionally, Graph Neural Networks (GNNs) research are normally separated into simple graphs and hypergraphs, and these two classes of methods tend not to interchange. Therefore, there is a need for a more flexible benchmark that allows GNNs to employ both simple edge and hyperedge information. In this paper, we present the *Semi-HyperGraph Benchmark (SHGB)*, a collection of comprehensive datasets combining hypergraphs and simple edges, with an accessible evaluation framework to fully understand the performance of GNNs on complex graphs. SHGB contains 23 real-world hypergraph datasets with simple edges included, across various domains such as biology, social media, and e-commerce. Furthermore, we provide an extensible evaluation framework and a supporting codebase to facilitate the training and evaluation of GNNs on SHGB. Our empirical study of existing GNNs on SHGB reveals various research opportunities and gaps, including (1) evaluating the actual performance improvement of hypergraph GNNs over simple graph GNNs; (2) comparing the impact of different sampling strategies on hypergraph learning methods; and (3) exploring ways to integrate simple edge and hyperedge information. We make our source code and full datasets publicly available at `https://anonymous-url/`.

## 1 INTRODUCTION

Graphs are powerful tools for capturing relationships between objects, ranging from social networks, biology (Rozemberczki et al., 2021) to e-commerce (McAuley et al., 2015; He & McAuley, 2016; Ni et al., 2019). However, real-world large-scale networks often involve complex node relations beyond only being pairwise. While hypergraphs (McCallum et al., 2000; Getoor, 2005; Sen et al., 2008; Chien et al., 2022) have been introduced to capture these complex node relations, they disregard the existence of simple edges, which introduces inflexibility to graph representation learning algorithms, and can lead to a drop in their performance. Although hypergraphs can include simple edges by using hyperedges with only two nodes, this is not a principled approach of utilising hyperedges, especially when they are employed to described clustering or homogeneity relations between multiple nodes, and can still potentially hurt machine learning systems' performance.

Existing Graph Neural Networks (GNNs) have been proposed for representation learning on simple graphs (Kipf & Welling, 2017; Hamilton et al., 2017; Veličković et al., 2018; Brody et al., 2022; Zeng et al., 2020) and hypergraphs (Feng et al., 2019; Yadati et al., 2019; Chan & Liang, 2020; Dong et al., 2020; Bai et al., 2021). However, they are usually evaluated separately either on simple graph datasets or hypergraph datasets, and tend not to interchange in terms of evaluation datasets. Besides, without a unified modeling of hypergraphs that employ simple edges, nor a collection of comprehensive datasets with an accessible evaluation framework, we cannot fully uncover the underlying performance of these GNNs on complex graphs.

To address these limitations, we endeavour to construct the *Semi-HyperGraph Benchmark (SHGB)*, with a more flexible view of hypergraphs including both datasets and an evaluation framework. Firstly, we extend the notion of hypergraphs by allowing simple edges to be included, rather than only hyperedges. This enables hypergraphs to capture more simple, pairwise node interactions, which not only increases flexibility for hypergraph learning algorithms, but also allows simple graph learning algorithms to be evaluated on hypergraphs. Based on this extension on hypergraphs, we build 23 real-world hypergraph datasets containing simple edges, across various domains such as biology, social media, and e-commerce. Compared with other existing hypergraph datasets (McCallum et al., 2000; Getoor, 2005; Sen et al., 2008; Chien et al., 2022) that are on a relatively small scale and do not facilitate simple edges, we believe our SHGB represents a significant step forward in the development of comprehensive and flexible datasets for evaluating complex graph learning algorithms.

Furthermore, to facilitate fair evaluation of GNNs on our proposed datasets, we provide an extensible evaluation framework and a supporting codebase. Our evaluation framework includes several common graph prediction tasks using their corresponding evaluation metrics, such as node classification and regression tasks. We benchmark seven widely-used GNN models (Kipf & Welling, 2017; Hamilton et al., 2017; Veličković et al., 2018; Brody et al., 2022; Zeng et al., 2020; Bai et al., 2021; Brody et al., 2022) on SHGB, and introduce three novel baselines that incorporate simple edge and hyperedge information, thereby enabling the researchers to conveniently evaluate their own models and compare the results. In addition, our empirical study of existing GNNs on SHGB reveals various research opportunities and gaps, which are elaborated in later sections.

To summarise, we provide a comprehensive and flexible framework for modelling and evaluating hypergraph learning methods with simple edges included, in the hope of stimulating further research in this field. We make the source code and complete datasets of SHGB publicly available. Our main contributions in this paper are as follows:

- We extend hypergraphs to a more flexible view by including simple edges, for fostering further study in representation learning on complex graphs.

- Inspired by this flexible framework, we construct the *Semi-HyperGraph Benchmark (SHGB)* consisting of 23 datasets covering a wide range of real-world applications. We then extend SHGB with an easy-to-use and extensible evaluation framework.

- Through extensive experimentation, we have verified both the necessity and superiority of our proposed datasets and benchmarking tool. We also draw insights for the graph representation learning community, such as (1) existing hypergraph GNNs may not keep outperforming simple graph GNNs on large-scale networks; (2) appropriate sampling strategies improve the performance of GNNs on hypergraphs; and (3) integrating simple edge and hyperedge information can significantly enhance the prediction performance on complex graphs.

## 2 RELATED WORK

**Graph Neural Networks for Simple Graphs**  Graph Neural Networks (GNNs) on simple graphs encode the nodes through neural networks, and learn the representations of the nodes through message-passing within the graph structure. GCNs (Kipf & Welling, 2017) incorporate the convolution operation into GNNs. GAT (Veličković et al., 2018) and GATv2 (Brody et al., 2022) are another family of GNN variants that improves the expressive power of GNNs through attention mechanisms. GraphSAGE (Hamilton et al., 2017) is a general inductive framework that leverages node information to efficiently generate node embeddings for previously unseen data. GraphSAINT (Zeng et al., 2020) emphasises the importance of graph sampling-based inductive learning method to improve training efficiency, especially for large graphs. While these models succeed in simple graph datasets, it is also of great research interest to test their performance on hypergraphs. However, there lacks a systematic evaluation of these models on hypergraph datasets.

**Graph Neural Networks for Hypergraphs**  Hypergraphs are designed to capture more complex node relations, where an edge can connect two or more nodes. In general, GNNs for hypergraphs optimise the node representation through a two-step process. Initially, the node embeddings within each hyperedge are aggregated to form a hidden embedding of each hyperedge. Subsequently, the hidden embeddings of hyperedges with common nodes are aggregated to update the representations

of their common nodes. Both HGNN (Feng et al., 2019) and HyperConv (Bai et al., 2021) precisely follow this process. The expressiveness of hypergraph GNNs could be enhanced by modifying this procedure. For instance, HyperGCN (Yadati et al., 2019) refines the node aggregation within hyperedges using mediators (Chan & Liang, 2020); HyperAtten (Bai et al., 2021) uses attention to measure the degree to which a node belongs to a hyperedge; HNHN (Dong et al., 2020) applies nonlinear functions to both node and edge aggregation processes; ED-HNN (Wang et al., 2023) approximates continuous equivariant hypergraph diffusion operators on hypergraphs by feeding node representations into the message from hyperedges to nodes.

**Existing Hypergraph Datasets** All above-mentioned hypergraph learning methods are evaluated on hypergraphs constructed from citation networks, such as Cora (McCallum et al., 2000), Cite-Seer (Getoor, 2005), PubMed (Sen et al., 2008) and DBLP[1]. Hyperedges are constructed using either of the two ways: co-citation (i.e., articles are grouped in the same hyperedge if they cite the same article) and co-authorship (i.e., articles are grouped in the same hyperedge if they share the same author). However, hypergraphs constructed from citation networks suffer from two severe disadvantages, making them inadequate evaluation metrics: (1) citation networks are too small in size and are prone to overfitting; and (2) the way hyperedges are constructed leads to substantial overlaps between hyperedges, thereby limiting their effectiveness in capturing multi-node relations. Chien et al. (2022) propose several more hypergraph datasets including adaptations of Yelp[2], Walmart (Amburg et al., 2020) and House (Chodrow et al., 2021), but these datasets are still relatively small-scale and have not been widely adopted by the hypergraph learning community.

In terms of how hyperedges are constructed, existing hypergraph datasets can be divided into two main categories: ground-truth based and rule based. While ground-truth based dataset do exist, they are usually limited in specific domains, and normally do not support the hyperedges to be constructed from a different perspective or a data modality than simple edges. The rule based hypergraph datasets also serve as an important part of the benchmarks and cannot be neglected (Feng et al., 2019).

## 3 DATASETS

### 3.1 EXTENDING HYPERGRAPHS WITH SIMPLE EDGES

Formally, a simple graph $\mathcal{G} = (\mathcal{V}, \mathcal{E})$ is a collection of nodes $\mathcal{V}$ and edges $\mathcal{E} \subseteq \mathcal{V} \times \mathcal{V}$ between pairs of nodes. This graph abstraction assumes that each edge only connects two nodes. However, as discussed in the previous sections, many real-world networks have more complex node relations than just pairwise relations. Hypergraphs attempt to capture such complex relations, by introducing hyperedges that can connect more than two nodes. A hypergraph $\mathcal{G} = (\mathcal{V}, \mathcal{E})$ is defined by a set of nodes $\mathcal{V}$ and a set of hyperedges $\mathcal{E}$, where hyperedges $e \in \mathcal{E}$ are arbitrary subsets of $\mathcal{V}$. Each hyperedge is also assigned a positive weight $w(e)$. The hypergraph modelling can indeed capture more complex node relations than a simple graph, but it overlooks the simple pairwise node relations provided by a simple graph. This makes hypergraphs lose substantial useful graph information in real-world applications, which normally contains both simple pairwise node relations and multi-node relations. Although it is possible to represent simple edges in hypergraphs using hyperedges with only two nodes, this is not a principled approach of utlising hyperedges. Treating such "simple" hyperedegs the same as genuine multi-node hyperedges—which normally describes local clustering and homogeneity information in a hypergraph—can potentially contaminate the hyperedge set, leading to a decline in hyperedge learning systems' training outcomes. Therefore, there is a need for a more flexible hypergraph abstraction that combines both simple edges and hyperegdes.

We extend hypergraphs to a "semi"-hypergraph style, by allowing it to include simple graphs. Formally, this extension can be defined as $\mathcal{G} = (\mathcal{V}, \mathcal{E}, \mathcal{E}_h)$, where $\mathcal{V}$ is the set of nodes, $\mathcal{E}$ is the set of simple edges, and $\mathcal{E}_h$ is the set of hyperedges. Each hyperedge $e_h \in \mathcal{E}_h$ is a non-empty subset of $\mathcal{V}$, which contains at least three nodes, and is assigned a positive weight $w(e_h)$. This flexible extension encloses simple graphs into the traditional definition of hypergraphs, allowing the representation of each specific graph type under the same view, by tightening relevant constraints:

---

[1] https://dblp.org/xml/release/
[2] https://www.yelp.com/dataset

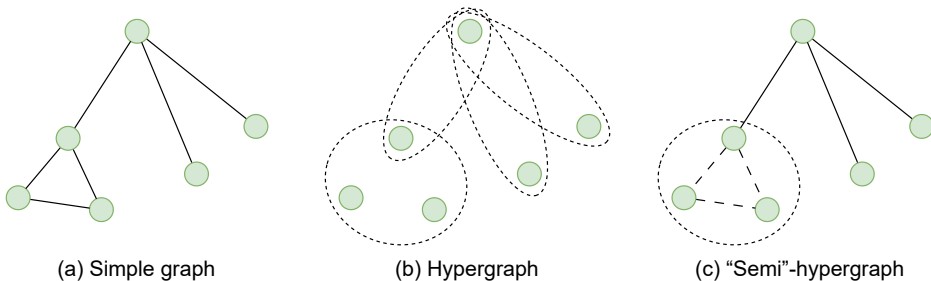

(a) Simple graph     (b) Hypergraph     (c) "Semi"-hypergraph

Figure 1: Modellings of different graph types. (a) Simple graphs contain only simple edges, which fails to capture hyperedge information; (b) Hypergraphs contain only hyperedges, which overlooks simple edge relations, and representing simple edges with hyperedges containing only two nodes is not a principled way for capturing simple edge information; (c) We extend hypergraphs to include simple edges, which increases their flexibility to capture both simple edge and hyperedge information.

- Simple graph: the semi-hypergraph extension represents a simple graph if and only if it only contains simple edges. The hyperedge set $\mathcal{E}_h$ is then empty and can be omitted.

- "General" hypergraph: the semi-hypergraph extension represents a general hypergraph if and only if it contains both simple edges and hyperedges. This is mathematically the same as a traditional hypergraph, except that the traditional hypergraph does not separate the simple edges $\mathcal{E}$ from the hyperedges set $\mathcal{E}_h$, and treat the simple edges as hyperedges containing only two nodes. This makes the traditional hypergraph less beneficial than the semi-hypergraph extension in practice.

- "Pure" hypergraph: the semi-hypergraph extension represents a "pure" hypergraph if and only if it only contains hyperedges. The simple edge set $\mathcal{E}$ is then empty and can be omitted.

Figure 1 illustrates the modellings of different graph types. In graph representation learning, a semi-hypergraph can be represented as a tuple of features $(\mathbf{X}, \mathbf{E}, \mathbf{A}, \mathbf{H}, \mathbf{W})$. $\mathbf{X} \in \mathbb{R}^{|\mathcal{V}| \times d_v}$ is the node feature matrix of the semi-hypergraph, with each row $\mathbf{x}_v \in \mathbb{R}^{d_v}$ being the $d_v$-dimensional features of node $v$. $\mathbf{E} \in \mathbb{R}^{(|\mathcal{E}|+|\mathcal{E}_h|) \times d_e}$ is the combined edge feature matrix of the semi-hypergraph, with each row $\mathbf{e}_e \in \mathbb{R}^{d_e}$ being the $d_e$-dimensional features of simple edge $e$ or hyperedge $e_h$. Sometimes the hyperedges do not have intrinsic embeddings, and their embeddings are either left blank or computed from the embeddings of the nodes they contain. $\mathbf{A} \in \{0, 1\}^{|\mathcal{V}| \times |\mathcal{V}|}$ is the simple edge adjacency matrix of the semi-hypergraph, where $A_{uv} = 1$ if $\{u, v\} \in \mathcal{E}$. $\mathbf{H} \in \{0, 1\}^{|\mathcal{V}| \times |\mathcal{E}_h|}$ is the node-hyperedge incidence matrix of the semi-hypergraph, with each entry $H_{ve_h} = 1$ if $v \in e_h$ and 0 otherwise. $\mathbf{W} \in \mathbb{R}^{|\mathcal{E}_h| \times |\mathcal{E}_h|}$ is a diagonal matrix containing the weights of the hyperedges, with each on-diagonal entry $W_{e_h e_h} = w(e_h)$. By performing $\mathbf{H}\mathbf{W}$, one can combine $\mathbf{H}$ and $\mathbf{W}$ as a single weighted hyperedge incidence matrix. However, as an initial setting, we choose to keep $\mathbf{H}$ and $\mathbf{W}$ separate, since it is the common practice adopted by almost all existing works on hypergraphs. By doing so, we allow other works to also follow the common practice, and potentially perform separate operations to these two matrices, or even adopt their own weight matrices for the hyperedges.

## 3.2 DATASET CONSTRUCTION

The semi-hypergraph, as a versatile data structure, presents a more comprehensive view of hypergraphs. It not only captures simple pairwise relationships as simple edges do, but also retains complex multi-node interactions in hyperedges. Therefore, it enhances traditional hypergraphs by capturing a broader range of data relationships. In light of this, we introduce the *Semi-HyperGraph Benchmark (SHGB)*, a novel collection of datasets consisting of 23 hypergraphs with simple edges included, derived from real-world networks across varied domains, including biology, social media, and e-commerce. Care has been taken to ensure that the datasets do not contain any personally identifiable information. Figure 2 provides an overview of these graphs and their construction mechanisms. Table 1 reports the key graph statistics for each dataset group. The SHGB datasets can be splitted into three groups according to their construction processes:

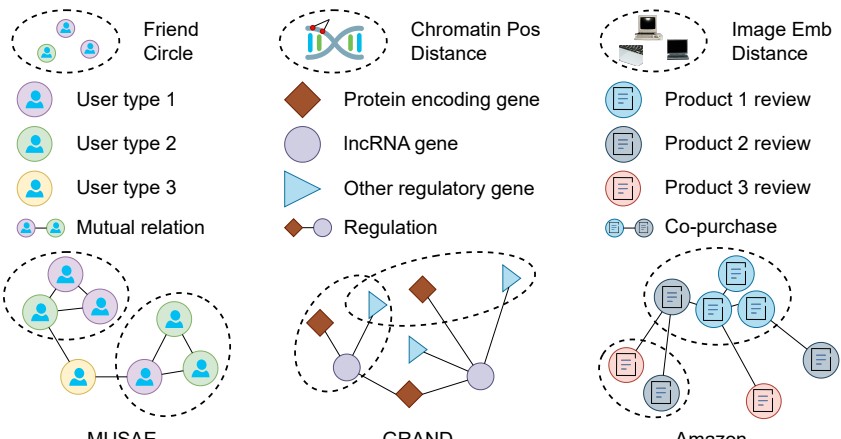

Figure 2: Construction of the SHGB datasets. The MUSAE datasets are social and knowledge networks, where the hyperedges are constructed from friend circles or mutually linked page groups. The GRAND datasets are gene regulatory networks, where the hyperedges are formed using the positions of genomic elements on the chromosome. The Amazon datasets are product co-review networks, where the hyperedges are built from the clusters of the image embeddings of the products.

**MUSAE**  We build eight social networks derived from the Facebook pages, GitHub developers and Twitch gamers, plus three English Wikipedia page-page networks on specific topics (chameleons, crocodiles and squirrels) based on the MUSAE (Rozemberczki et al., 2021) datasets. Nodes represent users or articles, and edges are mutual followers relationships between the users, or mutual links between the articles. In addition to the original MUSAE datasets, we construct the hyperedges to be mutually connected sub-groups that contain at least three nodes (i.e., maximal cliques with sizes of at least 3). We also enable each dataset to have an option to use either the raw node features, or the preprocessed node embeddings as introduced in MUSAE. The hyperedges in MUSAE serve as a complement to the simple edges, by precomputing the local clustering relations of the nodes. This can effectively save time and compute resources for any machine learning algorithms wishing to utilise hyperedge information from explicitly computing those hyperedges.

**GRAND**  We select and build ten gene regulatory networks in different tissues and diseases from GRAND (Ben Guebila et al., 2022), a public database for gene regulation. Nodes represent gene regulatory elements (Maston et al., 2006) with three distinct types: protein-encoding gene, lncRNA gene (Long et al., 2017), and other regulatory elements. Edges are regulatory effects between genes. The task is a multi-class classification of gene regulatory elements. We train a CNN (Eraslan et al., 2019) and use it to take the gene sequence as input and create a 4,651-dimensional embedding for each node. The hyperedges are constructed by grouping nearby genomic elements on the chromosomes, i.e., the genomic elements within 200k base pair distance are grouped as hyperedges. Since the geometric information of genes is not captured by the simple edges or node embeddings, the hyperedges in GRAND not only complement the simple edges, but also enhance the overall information.

**Amazon**  Following existing works on graph representation learning on e-commerce networks (Shchur et al., 2018; Zeng et al., 2020), we further build two e-commerce hypergraph datasets based on the Amazon Product Reviews dataset (McAuley et al., 2015; He & McAuley, 2016; Ni et al., 2019). Nodes represent products, and an edge between two products is established if a user buys these two products or writes reviews for both. However, unlike those existing datasets, we introduce the image modality into the construction of hyperedge. To be specific, the raw images are fed into a CLIP (Radford et al., 2021) classifier, and a 512-dimensional feature embedding for each image is returned to assist the clustering. The hyperedges are then constructed by grouping products whose image embeddings' pairwise distances are within a certain threshold. As the hyperedges in Amazon are constructed from a different modality than the simple edges and node embeddings, they provide both a complement to the simple edges and an enhancement to the overall information.

Table 1: Aggregated dataset statistics of SHGB.

| Name | #Graphs | Avg. #Nodes | Avg. #Edges | Avg. #Hyperedges | Avg. Node Degree | Avg. Hyperedge Degree | Avg. Clustering Coef. |
|---|---|---|---|---|---|---|---|
| MUSAE-GitHub | 1 | 37,700 | 578,006 | 223,672 | 30.7 | 4.6 | 0.168 |
| MUSAE-Facebook | 1 | 22,470 | 342,004 | 236,663 | 30.4 | 9.9 | 0.360 |
| MUSAE-Twitch | 6 | 5,686 | 143,038 | 110,142 | 50.6 | 6.0 | 0.210 |
| MUSAE-Wiki | 3 | 6,370 | 266,998 | 118,920 | 88.8 | 14.4 | 0.413 |
| GRAND-Tissues | 6 | 5,931 | 5,926 | 11,472 | 2.0 | 1.3 | 0.000 |
| GRAND-Diseases | 4 | 4,596 | 6,252 | 7,743 | 2.7 | 1.3 | 0.000 |
| Amazon-Computers | 1 | 10,226 | 55,324 | 10,226 | 10.8 | 4.0 | 0.249 |
| Amazon-Photos | 1 | 6,777 | 45,306 | 6,777 | 13.4 | 4.8 | 0.290 |

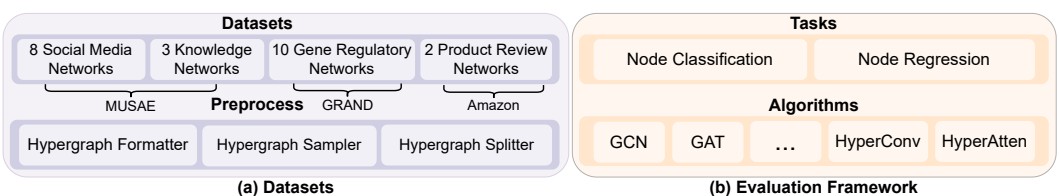

Figure 3: An overview of the SHGB framework. (a) The SHGB datasets include 23 real-world hypergraphs with simple edges included, which preserve both simple edge and hyperedge information. (b) We also build a lightweight evaluation framework using PyTorch Lightning and PyTorch Geometric.

While the hyperedges of the SHGB datasets are closer to the rule based category as discussed in Section 2, they provide more concrete rules that can be regarded as "almost" ground-truths, rather than just looking at the embedding space (MUSAE & GRAND), and allow the simple edges and hyperedges to be formed from different perspectives (GRAND) or modalities (Amazon).

## 4    EVALUATION FRAMEWORK

### 4.1    OVERVIEW

We create an extensible evaluation framework in SHGB, simplifying the process of training and assessing GNNs for both simple graphs and hypergraphs. Figure 3 illustrates the key components of SHGB. 23 semi-hypergraphs are used to train and evaluate the seven GNNs, including four GNNs for simple graphs: GCN (Kipf & Welling, 2017), GraphSAGE (Hamilton et al., 2017), GAT (Veličković et al., 2018), GATv2 (Brody et al., 2022); three hypergraph GNNs: HyperConv, HyperAtten (Bai et al., 2021), ED-HNN (Wang et al., 2023); and one sampling-based training strategy: GraphSAINT (Zeng et al., 2020). To fairly compare these models, we evaluate them under the same hyperparameter settings, which are listed in Appendix B. We repeat each experiment 5 times with different random seeds, and report their means and standard deviations in Appendix C. The experiments are evaluated with the accuracy for the node classification task and the mean square error (MSE) for node regression tasks. Our results highlight the challenges and research gaps in developing effective GNNs for real-world complex graphs:

- Existing hypergraph GNNs may not outperform simple graph GNNs, even when the hyperedges provide meaningful information to the task. (Section 4.2)

- The performance of conventional hypergraph GNNs can be improved with graph samplers that samples the node mainly based on the simple edge information. (Section 4.3)

- Combining both simple edge and hyperedge information can be substantially beneficial for node classification and regression tasks in hypergraphs. (Section 4.4)

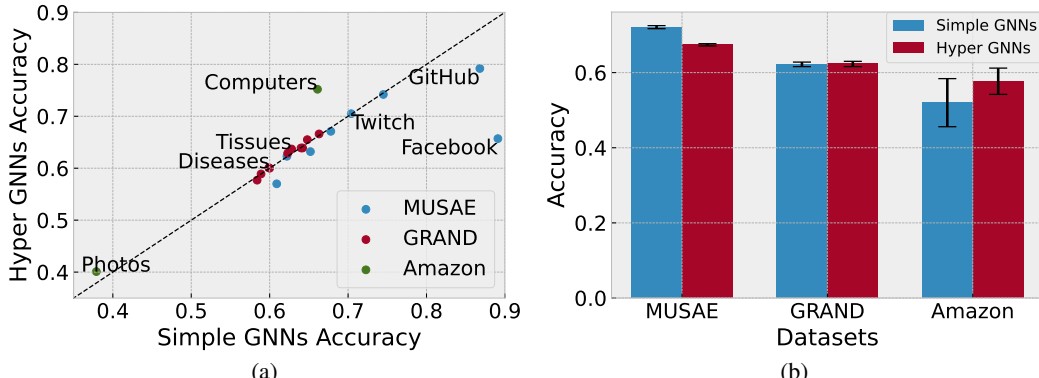

Figure 4: Evaluating the accuracy of simple-graph and hypergraph GNNs on SHGB: (a) The scatter plot of the accuracies of the best-performing hypergraph GNNs with respect to the best-performing simple graph GNNs on each dataset. The black dashed line at $y = x$ serves as a reference line where both GNN types perform equally well, while a dot above this reference line means hypergraph GNNs perform better than simple graph GNNs on a given node classification task, and vice versa. The plot indicates that the best-performing hypergraph GNNs match the best-performing simple graph GNNs' accuracies on GRAND, underperform on MUSAE, and outperform on Amazon. (b) The bar chart of the accuracies of both GNN types aggregated within MUSAE, GRAND, and Amazon, shows that the performance improvement of hypergraph GNNs on Amazon compared to simple graph GNNs may not be significant, after taking the standard deviations into consideration.

## 4.2 ASSESSING PERFORMANCE: HYPERGRAPH GNNS VS. SIMPLE GRAPH GNNS

We report the mean accuracies of four simple graph GNNs (GCN, GraphSAGE, GAT, GATv2) and three hypergraph GNNs (HyperConv, HyperAtten, ED-HNN) on SHGB in Appendix C. For MUSAE and GRAND, the information embedded in the hypergraph space and simple graph space do not contribute to the task objective. For the Amazon datasets, since the hyperedges are constructed using the embeddings of actual product images, nodes within a hyperedge can provide meaningful information to the task. Therefore, we expect hypergraph GNNs to outperform simple graph GNNs on Amazon. However, our experiments show that the performance gain from hypergraph GNNs on Amazon is only marginal.

Figure 4a shows a pairwise comparison of simple graph and hypergraph GNNs. In both MUSAE and GRAND, hypergraph GNNs perform equally or less well than the simple graph methods, while they perform better than simple graph GNNs on two Amazon Review graphs. However, Figure 4b, the aggregated mean accuracy across all graphs with the standard deviation, indicates that the performance gain of using hypergraph GNNs may not be significant.

## 4.3 OPTIMISED SAMPLING IN HYPERGRAPHS

Various hypergraph sampling strategies are proposed for sampling subgraphs with the purpose of preserving the graph statistics (Choe et al., 2022; Dyer et al., 2021). However, there is a lack of practical implementations of hypergraph samplers and evaluations of their efficacy. Following the work by GraphSAINT (Zeng et al., 2020) on simple graph sampling, we propose *HypergraphSAINT*, a class of hypergraph samplers employing GraphSAINT's graph sampling approaches. In HypergraphSAINT, we adopt the same sampling strategies in GraphSAINT for sampling the simple graph components in a hypergraph, making three different types of samplers: node sampler (HypergraphSAINT-Node), edge sampler (HypergraphSAINT-Edge), and random walk sampler (HypergraphSAINT-RW). As for the hyperedges, we use an intuitive procedure that any hyperedges containing at least one node in the sampled subgraph are retained, but all nodes not in the subgraph are masked out from those hyperedges. By using this method, we preserve the original hypergraph characteristics in the sampled subgraphs to the maximum extent. We also construct two naïve random samplers as baselines for evaluation: random node sampler and random hyperedge sampler, which randomly sample a subset of nodes/hyperedges from the original hypergraph according to a uniform sampling distribution.

Table 2: Statistics of the subgraphs obtained by various samplers.

| Samplers | MUSAE-GitHub | | | MUSAE-Facebook | | |
|---|---|---|---|---|---|---|
| | Avg. Node Degree | Avg. Hyperedge Degree | Clustering Coef. | Avg. Node Degree | Avg. Hyperedge Degree | Clustering Coef. |
| Original | *30.66* | *0.168* | *4.590* | *30.44* | *0.360* | *9.905* |
| HypergraphSAINT-Node | 42.59 | 0.268 | 13.20 | 42.72 | 0.268 | 13.11 |
| HypergraphSAINT-Edge | 38.63 | 0.366 | 11.42 | 38.66 | **0.366** | **11.50** |
| HypergraphSAINT-RW | **35.35** | **0.203** | **6.48** | **35.03** | 0.206 | 6.54 |
| RandomNode | 2.25 | 0.020 | 19.16 | 2.58 | 0.037 | 21.73 |
| RandomHyperedge | 72.38 | 0.282 | 7.48 | 72.74 | 0.279 | 7.44 |

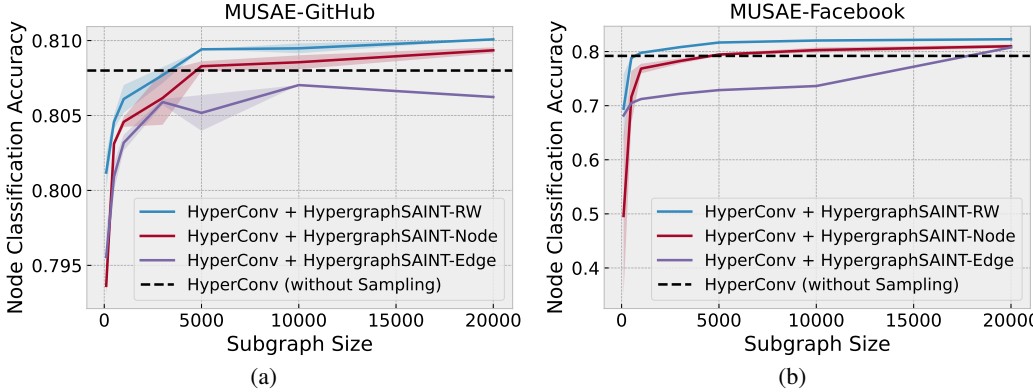

(a)  (b)

Figure 5: Node classification accuracy of different sampling techniques on (a) MUSAE-GitHub and (b) MUSAE-Facebook. The charts compare the accuracy of HyperConv model with three variants of HypergraphSAINT samplers at different subgraph sizes. The size of the subgraph is measured by the number of nodes. The black dashed line indicates the performance of HyperConv trained on the whole graph without any sampling. The accuracy increases as subgraph sizes increased, with both HypergraphSAINT-RW and HypergraphSAINT-Node outperforming standard HyperConv. HypergraphSAINT-RW consistently shows superior performance across different sampling sizes.

However, subgraphs sampled using the random node sampler can be very sparse, while subgraphs sampled using the random hyperedge sampler can be very dense.

We evaluate these samplers on MUSAE-GitHub and MUSAE-Facebook, the two largest graphs in SHGB. Firstly, we sample subgraphs with various samplers to examine how they preserve the structure of the original graphs. This is measured by three graph statistics: average node/hyperedge degree, and clustering coefficient. We sample subgraphs multiple times and report the average graph statistics in Table 2. Among all constructed samplers, HypergraphSAINT-Edge and HypergraphSAINT-RW perform the best in preserving the graph-level statistics. The two random samplers tend to sample subgraphs with distinct structures from the original hypergraphs.

We also evaluate HyperConv performance when paired with three different HypergraphSAINT samplers. Figure 5 shows distinct patterns regarding how the accuracy varied with different subgraph sizes. Evaluations were carried out on two distinct datasets, yielding the following key observations (1) There exists a positive correlation between the size of sampled subgraphs and model accuracy across both datasets. (2) At subgraph sizes of 5000 or larger, both HypergraphSAINT-RW and HypergraphSAINT-Node outperform vanilla HyperConv (trained on the whole graph without sampling). This highlights the potential for improved performance of hypergraph GNNs when integrating our HypergraphSAINT samplers. This performance gain could be explained by the reduction of over-smoothing brought by sampling subgraphs. We then apply these three HypergraphSAINT samplers to the 21 subsequent graphs in SHGB, extracting subgraphs of 1000, 2000, 3000, and 4000 nodes from each. The results of HyperConv with these three samplers on those graphs are summarised in Figure 7. Generally, the sampling techniques enhance HyperConv's accuracy across most graphs.

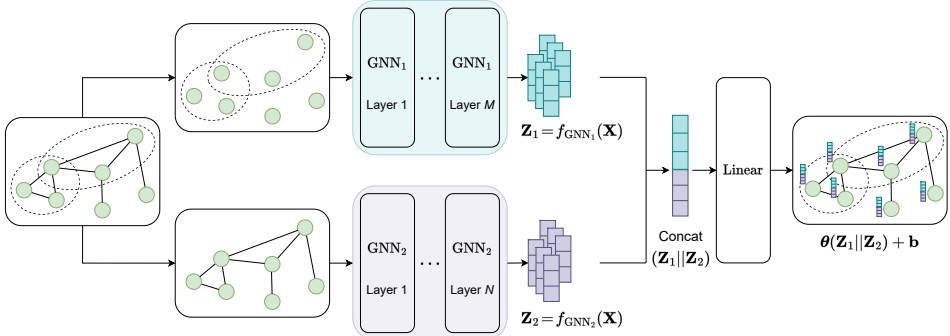

Figure 6: Architecture of the LP-GNN. A graph is fed into two GNNs, preferably a simple graph GNN and a hypergraph GNN, producing two sets of node embeddings. The two sets of node embeddings are then concatenated and pass through a linear layer to produce the final node embeddings.

Table 3: Accuracies of LP-GNNs, GCN, GAT, and HyperConv. The reciprocals of mean square errors (1/MSE) are reported for the GNNs' node regression task performances on the Chameleon dataset.

| Method | GitHub | Chameleon | Breast | Leukemia | Computers |
|---|---|---|---|---|---|
| RandomGuess | 0.250 | — | 0.333 | 0.333 | 0.100 |
| GCN | **0.872 ± 0.000** | 0.137 ± 0.000 | 0.639 ± 0.010 | 0.582 ± 0.001 | 0.756 ± 0.041 |
| GAT | 0.864 ± 0.001 | 0.152 ± 0.004 | 0.643 ± 0.001 | 0.587 ± 0.005 | 0.742 ± 0.043 |
| HyperConv | 0.808 ± 0.001 | 0.138 ± 0.000 | 0.645 ± 0.001 | 0.586 ± 0.003 | 0.842 ± 0.020 |
| LP-GCN+GAT | 0.867 ± 0.001 | 0.181 ± 0.009 | 0.626 ± 0.001 | 0.590 ± 0.002 | **0.930 ± 0.000** |
| LP-GCN+HyperConv | **0.872 ± 0.000** | 0.181 ± 0.001 | 0.652 ± 0.006 | **0.604 ± 0.004** | 0.913 ± 0.001 |
| LP-GAT+HyperConv | 0.860 ± 0.002 | **0.205 ± 0.002** | **0.657 ± 0.001** | 0.601 ± 0.002 | 0.930 ± 0.007 |

It's especially significant for regression tasks like MUSAE-Chameleon, MUSAE-Crocodile, and MUSAE-Squirrel, where the MSE improvement is considerable.

Overall, the HypergraphSAINT samplers, which mostly use simple edge information, can both effectively preserve hypergraph statistics and improve the performance of the hypergraph learning algorithms. This again underpins the advantage of combining hypergraphs and simple edges over conventional hypergraphs, emphasising the need to preserve both levels of information.

## 4.4 INTEGRATING SIMPLE GRAPH AND HYPERGRAPH INFORMATION

To test whether combining simple edge and hyperedge information could improve the performance of GNNs, we propose a simple algorithm called *Linear Probe Graph Neural Networks (LP-GNNs)*. LP-GNN consists of two GNNs $f_{\text{GNN}_1}$, $f_{\text{GNN}_2}$, plus a linear layer $f(\mathbf{x}) = \boldsymbol{\theta}\mathbf{x} + \mathbf{b}$. For a given semi-hypergraph $\mathcal{G} = (\mathbf{X}, \mathbf{E}, \mathbf{A}, \mathbf{H}, \mathbf{W})$, LP-GNN is defined as

$$\text{LP-GNN}(\mathbf{x}) = \boldsymbol{\theta} \cdot \text{CONCATENATE}(f_{\text{GNN}_1}(\mathbf{x}), f_{\text{GNN}_2}(\mathbf{x})) + \mathbf{b} \tag{1}$$

where $\mathbf{x}$ denotes the input features of a node. Figure 6 illustrates the architecture of an LP-GNN. For node regression tasks, LP-GNN($\mathbf{x}$) is directly used as the final output. For node classification tasks, a log softmax function is applied to LP-GNN($\mathbf{x}$) to produce the final output.

We evaluate LP-GNNs with the GNN pair being GCN+GAT, GCN+HyperConv, and GAT+HyperConv across all 23 SHGB datasets. The performance of these models are compared against those of the simple graph and hypergraph GNNs: GCN, GAT, and HyperConv. The main focus lies on the performance of LP-GCN+HyperConv and LP-GAT+HyperConv, which combine simple edge and hyperedge information. Notably, LP-GAT+HyperConv and LP-GCN+HyperConv surpass the other four methods in 18 of all 23 graphs in SHGB, which are summarised in Tables 12 to 17. Table 3 shows the performance of the three LP-GNNs on five selected SHGB datasets. As LP-GNNs employ a very simple strategy to combine information from two levels and show its advantage, they evident the potential advantages of integrating information across both levels.

## 5    CONCLUSION AND DISCUSSION

We introduce a useful and flexible extension of hypergraphs, by including simple edges to enhance hypergraph learning methods. Additionally, we present the *Semi-HyperGraph Benchmark (SHGB)*, a collection of real-world hypergraph datasets including simple edges, accompanied with an extensible GNN evaluation framework. Through extensive experiments, we demonstrate that existing hypergraph GNNs are not guaranteed to outperform simple graph GNNs on large-scale complex networks. In light of this, we propose a simple model called *Linear Probe Graph Neural Networks (LP-GNNs)*, which integrates simple edge and hyperedge information, and leverages the node prediction performance on the hypergraphs. We believe that SHGB can significantly aid the research in complex graph representation learning, and provide valuable insights to future research directions for the community, from the following two perspectives:

1. **Datasets and benchmarks:** we bring the notion of unifying the benchmarks for GNNs on different types of graphs (simple graphs, hypergraphs, and potentially other higher-order graphs), so that the graph representation learning community can use SHGB to conveniently and effectively evaluate whether a particular model designed to capture hyperedge information can indeed leverage the graph learning performance;

2. **Machine learning algorithms:**
   - With the simple model (LP-GNNs) integrating simple edge and hyperedge information that can already notably improve the node prediction performance, we would like to encourage the community to have more in-depth investigations on improving hypergraph learning performance by combining simple edge and hyperedge information;
   - We build the Amazon datasets where simple edges and hyperedges are constructing from different data modalities, in order to highlight the research insights of multimodal learning on graphs, and combining information from different modalities for improving graph learning performance.

**Limitations**    In MUSAE, hyperedges are constructed using the maximal cliques of a graph, which might lead to an overlap of information between the simple edges and hyperedges. In GRAND, the majority of gene regulatory graphs exhibit bipartite-like characteristics, which may be limited in connectivity for hyperedge constructions. In Amazon, hyperedges are created based on a preset threshold, and a more thorough investigation is required to find the range of the best threshold values.

**Future Work**    SHGB will be updated on a regular basis, and we welcome inputs from the community. In the future versions of SHGB, we plan to to enable multiple versions of Amazon datasets by adjusting various pairwise distance thresholds, and include more real-world large-scale networks. Additionally, as simply concatenating simple and hypergraph information followed by a linear transformation can already enhance the node prediction performance on hypergraphs, we will also seek to further improve the performance following this idea, by incorporating more fine-grained operations to combine simple graph and hypergraph information.

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

# A    FULL STATISTICS OF SHGB DATASETS

Table 4: Statistics of all 23 semi-hypergraph datasets in SHGB.

| Name | #Nodes | #Edges | #Hyperedges | Avg. Node Degree | Avg. Hyperedge Degree | #Node Features | #Classes |
|---|---|---|---|---|---|---|---|
| MUSAE-Github | 37,700 | 578,006 | 223,672 | 30.66 | 4.591 | 4,005 or 128 | 4 |
| MUSAE-Facebook | 22,470 | 342,004 | 236,663 | 30.44 | 9.905 | 4,714 or 128 | 4 |
| MUSAE-Twitch-DE | 9,498 | 306,276 | 297,315 | 64.49 | 7.661 | 3,170 or 128 | 2 |
| MUSAE-Twitch-EN | 7,126 | 70,648 | 13,248 | 19.83 | 3.666 | 3,170 or 128 | 2 |
| MUSAE-Twitch-ES | 4,648 | 118,764 | 77,135 | 51.10 | 5.826 | 3,170 or 128 | 2 |
| MUSAE-Twitch-FR | 6,549 | 225,332 | 172,653 | 68.81 | 5.920 | 3,170 or 128 | 2 |
| MUSAE-Twitch-PT | 1,912 | 62,598 | 74,830 | 65.48 | 7.933 | 3,170 or 128 | 2 |
| MUSAE-Twitch-RU | 4,385 | 74,608 | 25,673 | 34.03 | 4.813 | 3,170 or 128 | 2 |
| MUSAE-Wiki-Chameleon | 2,277 | 62,742 | 14,650 | 55.11 | 7.744 | 3,132 or 128 | Regression |
| MUSAE-Wiki-Crocodile | 11,631 | 341,546 | 121,431 | 58.73 | 4.761 | 13,183 or 128 | Regression |
| MUSAE-Wiki-Squirrel | 5,201 | 396,706 | 220,678 | 152.55 | 30.735 | 3,148 or 128 | Regression |
| GRAND-ArteryAorta | 5,848 | 5,823 | 11,368 | 1.991 | 1.277 | 4,651 | 3 |
| GRAND-ArteryCoronary | 5,755 | 5,722 | 11,222 | 1.989 | 1.273 | 4,651 | 3 |
| GRAND-Breast | 5,921 | 5,910 | 11,400 | 1.996 | 1.281 | 4,651 | 3 |
| GRAND-Brain | 6,196 | 6,245 | 11,878 | 2.016 | 1.296 | 4,651 | 3 |
| GRAND-Lung | 6,119 | 6,160 | 11,760 | 2.013 | 1.291 | 4,651 | 3 |
| GRAND-Stomach | 5,745 | 5,694 | 11,201 | 1.982 | 1.274 | 4,651 | 3 |
| GRAND-Leukemia | 4,651 | 6,362 | 7,812 | 2.736 | 1.324 | 4,651 | 3 |
| GRAND-Lungcancer | 4,896 | 6,995 | 8,179 | 2.857 | 1.334 | 4,651 | 3 |
| GRAND-Stomachcancer | 4,518 | 6,051 | 7,611 | 2.679 | 1.312 | 4,651 | 3 |
| GRAND-KidneyCancer | 4,319 | 5,599 | 7,369 | 2.593 | 1.297 | 4,651 | 3 |
| Amazon-Computers | 10,226 | 55,324 | 10,226 | 10.82 | 3.000 | 1,000 | 10 |
| Amazon-Photos | 6,777 | 45,306 | 6,777 | 13.37 | 4.800 | 1,000 | 10 |

# B    EXPERIMENTAL DETAILS

## B.1    TRAINING DETAILS

We run all the experiments on NVIDIA A100 PCIe GPU with 40GB RAM (Sulis) and NVIDIA V100 NVLink GPU with 32GB RAM (JADE), with each experiment taking less than 2 minutes. Adam (Kingma & Ba, 2015) is used as the optimiser, and CosineAnnealingLR (Gotmare et al., 2019) is used as the learning rate scheduler for all training. All models are trained for 50 epochs. For each experiment, the nodes of the used semi-hypergraph are split into the train, validation, and test sets with a split ratio of 6:2:2. For node classification tasks, BCEWithLogitsLoss is used as the loss function, which is defined as:

$$\mathcal{L}_{\text{BCEWithLogits}}(\mathbf{y}, \hat{\mathbf{y}}) = -\frac{1}{n} \sum_{i=1}^{n} \left[ y_i \cdot \log(\sigma(\hat{y}_i)) + (1 - y_i) \cdot \log(1 - \sigma(\hat{y}_i)) \right] \tag{2}$$

where $n$ is the total number of elements in $\mathbf{y}$ and $\hat{\mathbf{y}}$, $y_i$ is the $i$-th element of $\mathbf{y}$, the batch of true values, and $\hat{y}_i$ is the $i$-th element of $\hat{\mathbf{y}}$, the batch of raw (i.e., non-sigmoid-transformed) predicted values. $\sigma$ denotes the sigmoid function, which transforms the raw predictions into the range (0, 1). For node regression tasks, MSELoss is used as the loss function, which is defined as:

$$\mathcal{L}_{\text{MSE}}(\mathbf{y}, \hat{\mathbf{y}}) = \frac{1}{n} \sum_{i=1}^{n} (y_i - \hat{y}_i)^2 \tag{3}$$

where $n$ is the total number of elements in $\mathbf{y}$ and $\hat{\mathbf{y}}$, $y_i$ is the $i$-th element of $\mathbf{y}$, the batch of true values, and $\hat{y}_i$ is the $i$-th element of $\hat{\mathbf{y}}$, the batch of predicted values.

## B.2 HYPERPARAMETER SETTINGS

We perform a hyperparameter search for the learning rate and keep the hidden layer dimension the same for different models, the hyperparameters used for training each architecture are listed in Table 5. All seven GNNs (GCN, GraphSAGE, GAT, GATv2, HyperConv, HyperAtten, and GraphSAINT) share the same learning rate, hidden dimension, and dropout rate. HyperAtten has an additional hyperparameter, which is the hyperedge aggregation function. This function determines how the hyperedge is constructed from the nodes within it. The possible options for this function are 'sum' and 'concatenate'. In this work, we have selected 'sum' as the hyperedge aggregation function. For GraphSAINT, the additional hyperparameters are subgraph size, measured by the number of nodes in the subgraph, and the batch size, which is the number of subgraphs to sample in each epoch. Different subgraph sizes are applied according to the sizes of the original hypergraphs.

Table 5: Hyperparameter selections for the experiments.

| Method | Hyperparameter | Value |
|---|---|---|
| All | Learning rate | 0.01 |
| | Hidden dimension | 32 |
| | Dropout rate | 0.5 |
| HyperAtten | Hyperedge aggregation function | Sum |
| GraphSAINT | Subgraph size (MUSAE-GitHub, MUSAE-Facebook) | 5000 |
| | Subgraph size (MUSAE-Twitch-PT) | 1000 |
| | Subgraph size (others) | 3000 |
| | Batch size | 5 |

## C RESULTS

We evaluate the performance of eight GNNs on all 23 SHGB datasets. Each experiment is repeated five times with different random seeds, and the results are summarised in Tables 6 to 11. In the node classification tasks of MUSAE, GCN and GraphSAGE perform the best in the on GitHub and Facebook, as shown in Table 6, while ED-HNN performs the best on Twitch, as shown in Table 7. In the three node regression tasks on MUSAE-Wiki, GraphSAINT stands out among other GNNs, as shown in Table 11. Table 8 shows that ED-HNN and HyperAtten outperform other simple graph GNNs all six hypergraphs in GRAND-Tissues. For hypergraphs in GRAND-Diseases as shown in Table 9, GraphSAGE and ED-HNN exhibit superior performances. For the two Amazon hypergraph datasets, as shown in Table 10, ED-HNN consistently achieves the best performance. Overall, hypergraph GNNs tend to outperform simple graph GNNs on GRAND-Tissues and Amazon, perform equally as simple graph GNNs on MUSAE-Twitch, MUSAE-Wiki and GRAND-Diseases, and underperform simple graph GNNs on MUSAE-GitHub and MUSAE-Facebook.

Figure 7 summarises the performance of HyperConv (accuracy for prediction tasks, and MSE for regression tasks) with the three HypergraphSAINT samplers on 21 SHGB graphs other than MUSAE-GitHUB and MUSAE-Facebook, which are illustrated in Figure 5. The HypergraphSAINT sampling techniques generally enhance HyperConv's accuracy across most graphs, and are especially significant for regression tasks like MUSAE-Chameleon, MUSAE-Crocodile, and MUSAE-Squirrel.

Tables 12 to 17 reports the performances of LP-GNNs, GCN, GAT, and HyperConv on all 23 SHGB datasets. Notably, LP-GAT+HyperConv and LP-GCN+HyperConv surpass the other four methods in 18 of the 23 graphs. These results underscore the benefits of multi-level information integration in hypergraph representation learning.

Table 6: Accuracies of the selected GNNs on MUSAE-Facebook and GitHub datasets.

| Method | Facebook | GitHub |
|---|---|---|
| RandomGuess | 0.250 | 0.250 |
| GCN | $0.886 \pm 0.001$ | $\mathbf{0.872 \pm 0.000}$ |
| GraphSAGE | $\mathbf{0.902 \pm 0.002}$ | $0.871 \pm 0.002$ |
| GAT | $0.876 \pm 0.001$ | $0.864 \pm 0.001$ |
| GATv2 | $0.901 \pm 0.001$ | $0.866 \pm 0.001$ |
| HyperConv | $0.792 \pm 0.001$ | $0.808 \pm 0.001$ |
| HyperAtten | $0.523 \pm 0.002$ | $0.775 \pm 0.001$ |
| ED-HNN | $0.861 \pm 0.004$ | $0.862 \pm 0.001$ |
| GraphSAINT | $0.896 \pm 0.001$ | $0.871 \pm 0.001$ |

Table 7: Accuracies of the selected GNNs on the MUSAE-Twitch datasets.

| Method | TwitchES | TwitchFR | TwitchDE | TwitchEN | TwitchPT | TwitchRU |
|---|---|---|---|---|---|---|
| RandomGuess | 0.500 | 0.500 | 0.500 | 0.500 | 0.500 | 0.500 |
| GCN | $0.721 \pm 0.004$ | $0.624 \pm 0.001$ | $0.655 \pm 0.002$ | $\mathbf{0.620 \pm 0.003}$ | $0.689 \pm 0.006$ | $0.745 \pm 0.000$ |
| GraphSAGE | $0.690 \pm 0.002$ | $0.616 \pm 0.003$ | $0.657 \pm 0.001$ | $0.605 \pm 0.000$ | $0.672 \pm 0.013$ | $0.745 \pm 0.001$ |
| GAT | $0.694 \pm 0.002$ | $0.623 \pm 0.000$ | $0.645 \pm 0.004$ | $0.594 \pm 0.006$ | $0.664 \pm 0.007$ | $0.743 \pm 0.002$ |
| GATv2 | $0.710 \pm 0.003$ | $0.625 \pm 0.001$ | $0.651 \pm 0.003$ | $0.618 \pm 0.005$ | $0.687 \pm 0.009$ | $0.745 \pm 0.000$ |
| HyperConv | $0.715 \pm 0.001$ | $0.624 \pm 0.002$ | $0.654 \pm 0.002$ | $0.587 \pm 0.007$ | $\mathbf{0.701 \pm 0.005}$ | $0.741 \pm 0.001$ |
| HyperAtten | $0.695 \pm 0.000$ | $0.623 \pm 0.001$ | $0.610 \pm 0.003$ | $0.553 \pm 0.003$ | $0.641 \pm 0.000$ | $0.743 \pm 0.000$ |
| ED-HNN | $\mathbf{0.722 \pm 0.013}$ | $\mathbf{0.629 \pm 0.011}$ | $\mathbf{0.681 \pm 0.006}$ | $0.603 \pm 0.011$ | $0.695 \pm 0.015$ | $\mathbf{0.751 \pm 0.015}$ |
| GraphSAINT | $0.713 \pm 0.008$ | $0.622 \pm 0.003$ | $0.653 \pm 0.004$ | $0.610 \pm 0.011$ | $0.677 \pm 0.006$ | $0.746 \pm 0.002$ |

Table 8: Accuracies of the selected GNNs on the GRAND-Tissues datasets.

| Method | ArteryAorta | ArteryCoronary | Breast | Brain | Lung | Stomach |
|---|---|---|---|---|---|---|
| RandomGuess | 0.333 | 0.333 | 0.333 | 0.333 | 0.333 | 0.333 |
| GCN | $0.627 \pm 0.007$ | $0.662 \pm 0.001$ | $0.639 \pm 0.010$ | $0.625 \pm 0.000$ | $0.650 \pm 0.000$ | $0.643 \pm 0.000$ |
| GraphSAGE | $0.628 \pm 0.002$ | $0.663 \pm 0.001$ | $0.644 \pm 0.000$ | $0.618 \pm 0.002$ | $0.646 \pm 0.005$ | $0.630 \pm 0.010$ |
| GAT | $0.628 \pm 0.004$ | $0.663 \pm 0.000$ | $0.643 \pm 0.001$ | $0.625 \pm 0.001$ | $0.648 \pm 0.004$ | $0.643 \pm 0.000$ |
| GATv2 | $0.630 \pm 0.000$ | $0.663 \pm 0.000$ | $0.644 \pm 0.000$ | $0.624 \pm 0.001$ | $0.650 \pm 0.001$ | $0.642 \pm 0.000$ |
| HyperConv | $0.626 \pm 0.008$ | $0.662 \pm 0.000$ | $0.645 \pm 0.001$ | $0.625 \pm 0.000$ | $0.650 \pm 0.000$ | $0.643 \pm 0.000$ |
| HyperAtten | $0.647 \pm 0.003$ | $\mathbf{0.670 \pm 0.003}$ | $0.633 \pm 0.003$ | $0.632 \pm 0.003$ | $0.661 \pm 0.004$ | $0.636 \pm 0.004$ |
| ED-HNN | $\mathbf{0.667 \pm 0.013}$ | $0.667 \pm 0.015$ | $\mathbf{0.661 \pm 0.014}$ | $\mathbf{0.663 \pm 0.013}$ | $\mathbf{0.670 \pm 0.010}$ | $\mathbf{0.661 \pm 0.006}$ |
| GraphSAINT | $0.630 \pm 0.000$ | $0.663 \pm 0.000$ | $0.644 \pm 0.000$ | $0.625 \pm 0.000$ | $0.650 \pm 0.000$ | $0.643 \pm 0.000$ |

Table 9: Accuracies of the selected GNNs on the GRAND-Diseases datasets.

| Method | Leukemia | LungCancer | StomachCancer | KidneyCancer |
|---|---|---|---|---|
| RandomGuess | 0.333 | 0.333 | 0.333 | 0.333 |
| GCN | $0.582 \pm 0.001$ | $0.596 \pm 0.001$ | $0.602 \pm 0.007$ | $0.581 \pm 0.002$ |
| GraphSAGE | $\mathbf{0.604 \pm 0.016}$ | $\mathbf{0.615 \pm 0.015}$ | $0.602 \pm 0.016$ | $0.596 \pm 0.014$ |
| GAT | $0.587 \pm 0.005$ | $0.596 \pm 0.000$ | $0.600 \pm 0.007$ | $0.581 \pm 0.003$ |
| GATv2 | $0.583 \pm 0.000$ | $0.591 \pm 0.005$ | $0.596 \pm 0.002$ | $0.579 \pm 0.006$ |
| HyperConv | $0.586 \pm 0.003$ | $0.593 \pm 0.003$ | $0.596 \pm 0.004$ | $0.577 \pm 0.006$ |
| HyperAtten | $0.593 \pm 0.008$ | $0.608 \pm 0.008$ | $0.604 \pm 0.022$ | $0.578 \pm 0.012$ |
| ED-HNN | $0.603 \pm 0.013$ | $0.602 \pm 0.014$ | $\mathbf{0.610 \pm 0.011}$ | $\mathbf{0.603 \pm 0.011}$ |
| GraphSAINT | $0.583 \pm 0.000$ | $0.595 \pm 0.000$ | $0.596 \pm 0.000$ | $0.582 \pm 0.000$ |

Table 10: Accuracies of the selected GNNs on the Amazon datasets.

| Method | Computers | Photos |
|---|---|---|
| RandomGuess | 0.100 | 0.100 |
| GCN | $0.756 \pm 0.041$ | $0.295 \pm 0.017$ |
| GraphSAGE | $0.582 \pm 0.108$ | $0.366 \pm 0.061$ |
| GAT | $0.742 \pm 0.043$ | $0.434 \pm 0.074$ |
| GATv2 | $0.566 \pm 0.046$ | $0.420 \pm 0.075$ |
| HyperConv | $0.842 \pm 0.020$ | $0.337 \pm 0.059$ |
| HyperAtten | $0.663 \pm 0.005$ | $0.465 \pm 0.033$ |
| ED-HNN | $\mathbf{0.973 \pm 0.002}$ | $\mathbf{0.786 \pm 0.012}$ |
| GraphSAINT | $0.875 \pm 0.020$ | $0.512 \pm 0.141$ |

Table 11: MSEs ($\downarrow$) of the selected GNNs on the MUSAE-Wiki datasets.

| Method | Chameleon | Squirrel | Crocodile |
|---|---|---|---|
| GCN | $7.319 \pm 0.000$ | $8.761 \pm 0.001$ | $6.779 \pm 0.005$ |
| GraphSAGE | $6.945 \pm 0.005$ | $8.310 \pm 0.003$ | $6.380 \pm 0.005$ |
| GAT | $6.557 \pm 0.154$ | $8.093 \pm 0.054$ | $6.249 \pm 0.261$ |
| GATv2 | $7.290 \pm 0.019$ | $8.600 \pm 0.011$ | $6.717 \pm 0.005$ |
| HyperConv | $7.230 \pm 0.002$ | $8.706 \pm 0.000$ | $6.712 \pm 0.001$ |
| HyperAtten | $7.451 \pm 0.000$ | $8.782 \pm 0.000$ | $6.942 \pm 0.000$ |
| GraphSAINT | $\mathbf{5.165 \pm 0.027}$ | $\mathbf{7.541 \pm 0.023}$ | $\mathbf{4.898 \pm 0.035}$ |

Table 12: Accuracies LP-GNNs and other baselines on MUSAE-Facebook and GitHub.

| Method | Facebook | GitHub |
|---|---|---|
| RandomGuess | 0.250 | 0.250 |
| GCN | $0.886 \pm 0.001$ | $0.872 \pm 0.000$ |
| GAT | $0.876 \pm 0.001$ | $0.864 \pm 0.001$ |
| HyperConv | $0.792 \pm 0.001$ | $0.808 \pm 0.001$ |
| LP-GCN+GAT | $\mathbf{0.910 \pm 0.001}$ | $0.867 \pm 0.001$ |
| LP-GCN+HyperConv | $0.898 \pm 0.000$ | $\mathbf{0.872 \pm 0.000}$ |
| LP-GAT+HyperConv | $0.905 \pm 0.000$ | $0.860 \pm 0.002$ |

Table 13: Accuracies LP-GNNs and other baselines on MUSAE-Twitch.

| Method | TwitchES | TwitchFR | TwitchDE | TwitchEN | TwitchPT | TwitchRU |
|---|---|---|---|---|---|---|
| RandomGuess | 0.500 | 0.500 | 0.500 | 0.500 | 0.500 | 0.500 |
| GCN | $0.721 \pm 0.004$ | $0.624 \pm 0.001$ | $0.655 \pm 0.002$ | $\mathbf{0.620 \pm 0.003}$ | $0.689 \pm 0.006$ | $\mathbf{0.745 \pm 0.000}$ |
| GAT | $0.694 \pm 0.002$ | $0.623 \pm 0.000$ | $0.645 \pm 0.004$ | $0.594 \pm 0.006$ | $0.664 \pm 0.007$ | $0.743 \pm 0.002$ |
| HyperConv | $0.715 \pm 0.001$ | $0.624 \pm 0.002$ | $0.654 \pm 0.002$ | $0.587 \pm 0.007$ | $0.701 \pm 0.005$ | $0.741 \pm 0.001$ |
| LP-GCN+GAT | $0.727 \pm 0.001$ | $0.623 \pm 0.001$ | $\mathbf{0.662 \pm 0.001}$ | $0.612 \pm 0.002$ | $0.686 \pm 0.008$ | $0.745 \pm 0.001$ |
| LP-GCN+HyperConv | $\mathbf{0.729 \pm 0.001}$ | $\mathbf{0.626 \pm 0.000}$ | $0.657 \pm 0.001$ | $0.607 \pm 0.001$ | $\mathbf{0.696 \pm 0.004}$ | $0.744 \pm 0.000$ |
| LP-GAT+HyperConv | $0.714 \pm 0.001$ | $0.622 \pm 0.000$ | $0.654 \pm 0.001$ | $0.608 \pm 0.003$ | $0.672 \pm 0.003$ | $0.744 \pm 0.000$ |

Table 14: Accuracies of LP-GNNs and other baselines on GRAND-Tissues.

| Method | ArteryAorta | ArteryCoronary | Breast | Brain | Lung | Stomach |
|---|---|---|---|---|---|---|
| RandomGuess | 0.333 | 0.333 | 0.333 | 0.333 | 0.333 | 0.333 |
| GCN | $0.627 \pm 0.007$ | $0.662 \pm 0.001$ | $0.639 \pm 0.010$ | $0.625 \pm 0.000$ | $0.650 \pm 0.000$ | $0.643 \pm 0.000$ |
| GAT | $0.628 \pm 0.004$ | $0.663 \pm 0.000$ | $0.643 \pm 0.001$ | $0.625 \pm 0.001$ | $0.648 \pm 0.004$ | $0.643 \pm 0.000$ |
| HyperConv | $0.626 \pm 0.008$ | $0.662 \pm 0.000$ | $0.645 \pm 0.001$ | $0.625 \pm 0.000$ | $0.650 \pm 0.000$ | $0.643 \pm 0.000$ |
| LP-GCN+GAT | $0.627 \pm 0.000$ | $0.641 \pm 0.000$ | $0.626 \pm 0.001$ | $0.625 \pm 0.003$ | $0.627 \pm 0.001$ | $0.626 \pm 0.001$ |
| LP-GCN+HyperConv | $0.647 \pm 0.003$ | $0.660 \pm 0.003$ | $0.652 \pm 0.006$ | $0.637 \pm 0.002$ | $0.654 \pm 0.004$ | $0.654 \pm 0.001$ |
| LP-GAT+HyperConv | $\mathbf{0.649 \pm 0.001}$ | $\mathbf{0.664 \pm 0.001}$ | $\mathbf{0.657 \pm 0.001}$ | $\mathbf{0.645 \pm 0.001}$ | $\mathbf{0.662 \pm 0.002}$ | $\mathbf{0.656 \pm 0.002}$ |

Table 15: Accuracies of LP-GNNs and other baselines on GRAND-Diseases.

| Method | Leukemia | LungCancer | StomachCancer | KidneyCancer |
|---|---|---|---|---|
| RandomGuess | 0.333 | 0.333 | 0.333 | 0.333 |
| GCN | $0.582 \pm 0.001$ | $0.596 \pm 0.001$ | $0.602 \pm 0.007$ | $0.581 \pm 0.002$ |
| GAT | $0.587 \pm 0.005$ | $0.596 \pm 0.000$ | $0.600 \pm 0.007$ | $0.581 \pm 0.003$ |
| HyperConv | $0.586 \pm 0.003$ | $0.593 \pm 0.003$ | $0.596 \pm 0.004$ | $0.577 \pm 0.006$ |
| LP-GCN+GAT | $0.590 \pm 0.002$ | $0.583 \pm 0.002$ | $0.581 \pm 0.004$ | $0.584 \pm 0.002$ |
| LP-GCN+HyperConv | $\mathbf{0.604 \pm 0.004}$ | $0.614 \pm 0.006$ | $0.605 \pm 0.003$ | $0.609 \pm 0.004$ |
| LP-GAT+HyperConv | $0.601 \pm 0.002$ | $\mathbf{0.618 \pm 0.003}$ | $\mathbf{0.621 \pm 0.002}$ | $\mathbf{0.621 \pm 0.002}$ |

Table 16: Accuracies of LP-GNNs and other baselines on Amazon.

| Method | Computers | Photos |
|---|---|---|
| RandomGuess | 0.1 | 0.1 |
| GCN | $0.756 \pm 0.041$ | $0.295 \pm 0.017$ |
| GAT | $0.742 \pm 0.043$ | $0.434 \pm 0.074$ |
| HyperConv | $0.842 \pm 0.020$ | $0.337 \pm 0.059$ |
| LP-GCN+GAT | $\mathbf{0.930 \pm 0.000}$ | $0.711 \pm 0.005$ |
| LP-GCN+HyperConv | $0.913 \pm 0.001$ | $0.698 \pm 0.005$ |
| LP-GAT+HyperConv | $\mathbf{0.930 \pm 0.007}$ | $\mathbf{0.715 \pm 0.003}$ |

Table 17: MSEs ($\downarrow$) of LP-GNNs and other baselines on MUSAE-Wiki.

| Method | Squirrel | Crocodile | Chameleon |
|---|---|---|---|
| GCN | $7.319 \pm 0.000$ | $8.761 \pm 0.001$ | $6.779 \pm 0.005$ |
| GAT | $6.557 \pm 0.154$ | $8.093 \pm 0.054$ | $6.249 \pm 0.261$ |
| HyperConv | $7.230 \pm 0.002$ | $8.706 \pm 0.000$ | $6.712 \pm 0.001$ |
| LP-GCN+GAT | $7.554 \pm 0.029$ | $\mathbf{4.827 \pm 0.087}$ | $5.203 \pm 0.147$ |
| LP-GCN+HyperConv | $7.313 \pm 0.007$ | $4.851 \pm 0.014$ | $5.515 \pm 0.008$ |
| LP-GAT+HyperConv | $\mathbf{6.049 \pm 0.204}$ | $4.875 \pm 0.031$ | $\mathbf{4.665 \pm 0.050}$ |

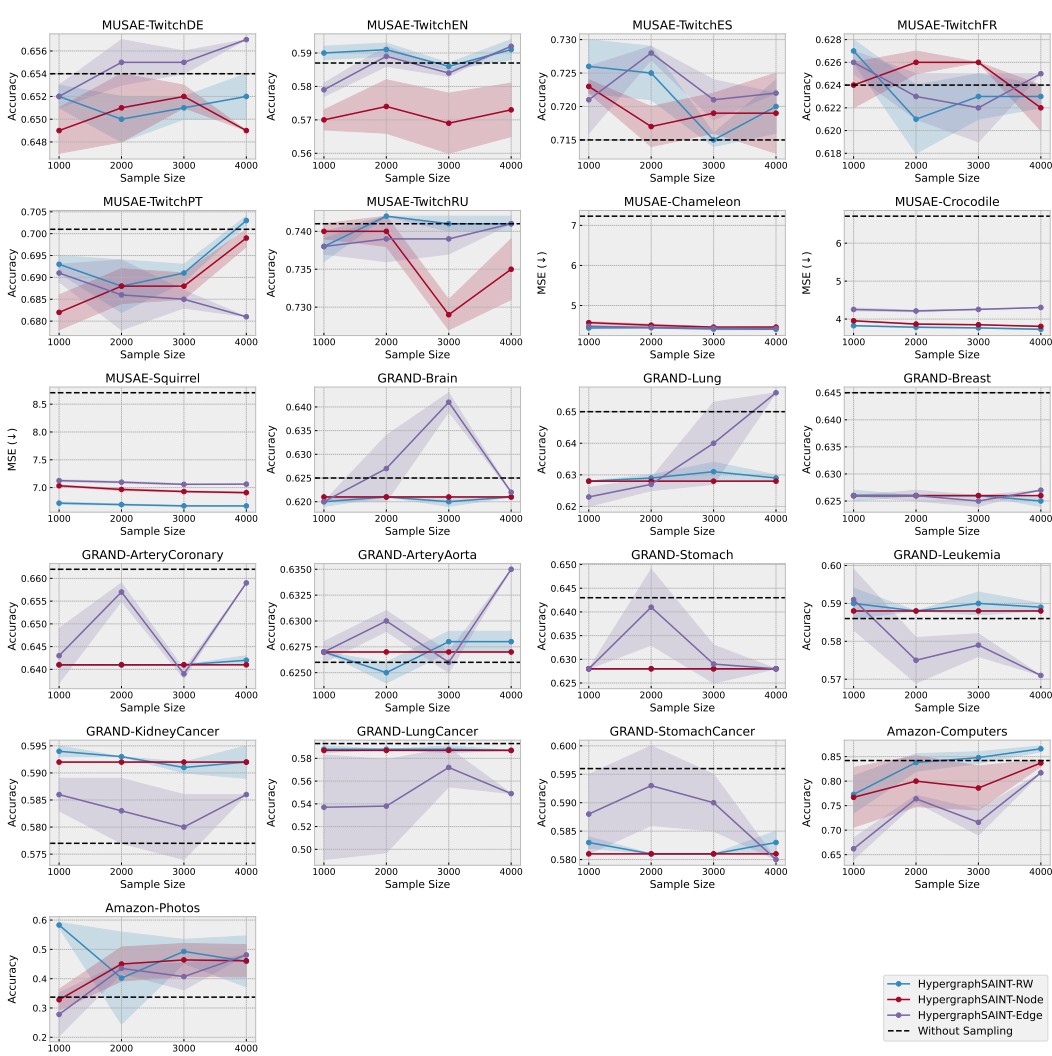

Figure 7: Node prediction performances of different sampling techniques on 21 SHGB datasets.

# D    DATA ACCESSIBILITY

The source code and full datasets of SHGB is publicly available at `https://anonymous-url/`. While the raw dataset in JSON format is hosted at `https://anonymous-url/`, we recommend the users to access the datasets through our Python library `anonymous-library`, which is installable via `pip`. This would allow the users to read the semi-hypergraphs in the format of PyTorch Geometric `Data` objects.

# E    LICENCE

The raw data for the MUSAE datasets are licenced under the the GNU General Public Licence, version 3 (GPLv3)[3]. The raw data for the GRAND datasets are licenced under the Creative Commons Attribution-ShareAlike 4.0 International Public Licence (CC BY-SA 4.0)[4]. The raw data for the Amazon datasets are licenced under the Amazon Service licence[5]. Having carefully observed the licence requirements of all data sources and code dependencies, we apply the following licence to our source code and datasets:

- The source code of SHGB is licenced under the MIT licence[6];
- The MUSAE and GRAND datasets are licenced under the GPLv3 licence[3];
- The Amazon datasets are licenced under the Amazon Service licence[5].

# F    ETHICS STATEMENT

All datasets constructed in SHGB are generated from public open-source datasets, and the original raw data downloaded from the data sources do not contain any personally identifiable information or other sensitive contents. The node prediction tasks for the SHGB datasets are designed to ensure that they do not, by any means, lead to discriminations against any social groups. Therefore, we are not aware of any social or ethical concern of SHGB. Since SHGB is a general benchmarking tool for representation learning on complex graphs, we also do not forsee any direct application of SHGB to malicious purposes. However, the users of SHGB should be aware of any potential negative social and ethical impacts that may arise from their chosen downstream datasets or tasks outside of SHGB, if they intend to use the SHGB datasets as pre-training datasets to perform trasnfer learning.

---

[3] `https://www.gnu.org/licenses/gpl-3.0.html`
[4] `https://creativecommons.org/licenses/by-sa/4.0/`
[5] `https://s3.amazonaws.com/amazon-reviews-pds/LICENSE.txt`
[6] `https://opensource.org/license/mit/`

