# OpenReview forum: "Semi-HyperGraph Benchmark: Enhancing Flexibility of Hypergraph Learning with Datasets and Benchmarks"
_ICLR.cc/2024/Conference — ICLR 2024 Conference Withdrawn Submission_

### Official Review · Reviewer_Smxv · 2023-10-17

**Soundness:** 2 fair
**Presentation:** 2 fair
**Contribution:** 2 fair
**Rating:** 5
**Confidence:** 4

**Summary:**

The paper introduces the Semi-HyperGraph Benchmark (SHGB) to unify graph and hypergraph benchmarks, attempting to bridge an existing gap where these benchmarks are considered separately in the literature.

SHGB integrates real-world datasets featuring both pairwise edges and hyperedges, enabling researchers to thoroughly assess Graph Neural Networks (GNNs) using a combination of edges and hyperedges.

Experiments reveal that
* GNNs on hypergraphs may not consistently outperform simple graph GNNs on large networks,
* sampling strategies enhance GNN performance on hypergraphs, and
* combining edge and hyperedge information improves predictions on complex graphs.

**Strengths:**

### Clarity
1. The paper provides a clear demonstration of edge and, more crucially, hyperedge construction through clear illustrations in Figures 1 and 2.
2. The SHGB framework is effectively outlined in Figure 3, ensuring a clear and concise understanding of its structure and components.
3. The paper is well-structured into clear sections, guiding readers logically from the introduction to the conclusion, ensuring a thorough understanding of the research content.

**Weaknesses:**

### Originality
1. The hyperedges in the datasets are *not natural* but derived from social networks' simple edges, a specified base pair distance in gene data, and e-commerce product image embeddings, which can be modelled as node features.
2. The hyperedges introduced can be obtained solely from *appropriately chosen graph data and node features*, limiting the originality of the curated hyperedges.
3. The HypergraphSAINT sampler is a straightforward application of GraphSAINT [Zeng et al., 2020] to hyperedges.

$~$
### Significance
4. Appropriately chosen GNNs (e.g., subgraph GNNs, MixHop, JKNets) on meticulously chosen graph datasets would be able to model the information given by the curated hyperedges, limiting the benchmark's potential impact.
5. The significance of the work can be improved by the inclusion of recent hypergraph neural networks [e.g., Chien et al., 2022, Wang et al., 2023] that generate hyperedge embeddings to fully exploit hyperedges.

$~$
### Quality
6. Contrary to the stated claim of extending hypergraphs with simple edges, e.g., see contribution 1 on page 2, the actual contribution involves integrating *curated hyperedges* with *naturally occurring edges* such as mutual followers in social networks and regulatory effects between genes in biological networks.
7. Four different GNNs on pairwise edges are tested in the experiments, whereas the selection of GNNs on hyperedges is restricted to those introduced in a single paper [Bai et al., 2021], further emphasising simple edges over hyperedges.

\
References:
* [Wang et al., 2023]: Equivariant Hypergraph Diffusion Neural Operators, ICLR'23
* [Chien et al., 2022]: You are AllSet: A Multiset Function Framework for Hypergraph Neural Networks, ICLR'22
* [Bai et al., 2021]: Hypergraph convolution and hypergraph attention, Pattern Recognition'21
* [Zeng et al., 2020]: GraphSAINT: Graph Sampling Based Inductive Learning Method, ICLR'20

**Questions:**

1. From a pure dataset perspective, what unique information do the hyperedges contribute that is not already represented by carefully/appropriately/meticulously chosen graph data and node features?
2. From a pure dataset perspective, are there unique insights into the specific reasons for choosing social networks' friend circles from pairwise edges, gene data within a user-specified base pair distance, and e-commerce product image embeddings as sources for deriving hyperedges?
3.  Have there been examples or scenarios where such derived hyperedges have demonstrated real-world applicability or have been utilised successfully in practical applications?
4. Considering the potential benefits of incorporating recent hypergraph neural networks [e.g., Chien et al., 2022, Wang et al., 2023], are there any challenges or limitations associated with their implementation, and if so, how could these challenges be addressed to ensure seamless integration into the proposed SHGB framework?
5. In what ways do the hyperedges complement or enhance the information captured by naturally occurring edges, such as mutual followers in social networks and regulatory effects between genes in biological networks?
6. In addition to Table 3, are there specific examples or case studies where this integration has led to unique insights or improved predictions compared to utilising simple edges alone?
7. Why was the selection of GNNs for hyperedges limited to those introduced in a single paper [Bai et al., 2021]?
8. Were there specific criteria or reasons for this restriction, and how does this choice impact the overall diversity and representation of GNN models applied to hyperedges?

---

> ### Author Response · Authors · 2023-11-22
> **Response to Reviewer Smxv (1)**
>
> Thank you for your detailed review and insightful comments. Please kindly see below for our responses to your comments:
>
> ### 1. From a pure dataset perspective, what unique information do the hyperedges contribute that is not already represented by carefully/appropriately/meticulously chosen graph data and node features?
>
> The hyperedges contribute unique information for complex graphs from the following two perspectives:
>
> 1. **Hyperedges can capture local clustering and homophilic properties of complex graphs more directly than simple edges.** As a simple edge can only connect two nodes, additional pre-computations are required to detect local clustering relations between multiple nodes if only simple edges are present, which could inevitably lead to a higher demand on compute resources and potentially hurt the training performance. In the MUSAE datasets of SHGB, we have pre-computed the hyperedges so that both works on simple graph GNNs and hypergraph GNNs can benefit from these datasets, and can be evaluated on a unified benchmark.
> 2. **Hyperedges can contribute information about complex graphs from another perspective or even another data modality.** In the GRAND datasets of SHGB, hyperedges are constructed based on the geometric information of genes, while the node embeddings are constructed from the gene sequences, and the simple edges represent gene regulatory effects. In the Amazon datasets of SHGB, hyperedges are generated from another modality (image) from the node embeddings (text), whereas simple edges represent ground-truths about co-reviews and co-purchases. We have shown in Tables 3 and 12-17 that combining information from different perspectives or modalities can significantly improve the learning performance than only learning one of the two information views. In addition, if the hyperedges that come from a different modality are naively encoded into the node embeddings, it would create extra difficulties for a GNN to thoroughly learn both information, thereby potentially limiting its performance.
>
> ### 2. From a pure dataset perspective, are there unique insights into the specific reasons for choosing social networks' friend circles from pairwise edges, gene data within a user-specified base pair distance, and e-commerce product image embeddings as sources for deriving hyperedges?
>
> The hyperedges are constructed in three diverse processes in the three groups of our SHGB, which covers a wide range of unique insights for complex graph learning:
>
> - In the MUSAE datasets of SHGB, we construct hyperedges by pre-computing the clustering information from simple edges, which create conveniences for both simple graph GNN and hypergraph GNN works, and allow them to be evaluated under a unified benchmark.
> - In the GRAND datasets of SHGB, in addition to the simple edges that represent gene regulatory effects, we construct hyperedges from another perspective, which is the geometric information of the genes. Geometric information of the genes are unique and critical in gene regulatory element predictions and cannot be neglected. In addition, building geometric information into hyperedges can capture multi-node clustering information of the genes more directly then modelling them into positional encodings in individual node embeddings.
> - In the Amazon datasets of SHGB, the hyperedges are constructed from another data modality (image) from the node embeddings (text), which encourages the community to combine those information to perform multimodal learning on complex graphs.
>
> ### 3. Have there been examples or scenarios where such derived hyperedges have demonstrated real-world applicability or have been utilised successfully in practical applications?
>
> We have demonstrated in Section 4.2, Figure 4 and Appendix C of our paper that hypergraph GNNs perform better than simple graph GNNs in the Amazon datasets, equally in the GRAND datasets, and underperform in the MUSAE datasets. Therefore, we have shown that in the Amazon datasets, our derived hyperedge information contributes more to the prediction tasks than the simple edges and have already been utilised successfully. We would also like to use MUSAE and GRAND to challenge the hypergraph learning community, where the success of the existing works cannot be fully generalised to other domains and on a larger scale.
>
> Furthermore, we have demonstrated with our simple baseline model (LP-GNNs) which integrate simple edge and hyperedge information and can already improve the node prediction performance than pure simple graph GNNs and pure hypergraph GNNs. This shows that the our derived hyperedge information can be successfully utilised by combining with simple edge information, and highlights a great potential for future research.
>
> (To be continued in the next response)

---

> ### Author Response · Authors · 2023-11-22
> **Response to Reviewer Smxv (2)**
>
> (Continued from the previous response)
>
> ### 4&7&8. Considering the potential benefits of incorporating recent hypergraph neural networks [e.g., Chien et al., 2022, Wang et al., 2023], are there any challenges or limitations associated with their implementation, and if so, how could these challenges be addressed to ensure seamless integration into the proposed SHGB framework? Why was the selection of GNNs for hyperedges limited to those introduced in a single paper [Bai et al., 2021]? Were there specific criteria or reasons for this restriction, and how does this choice impact the overall diversity and representation of GNN models applied to hyperedges?
>
> Thank you very much for pointing out the more recent hypergraph GNNs in addition to the models that we have originally included [Bai et al., 2021]. We originally selected [Bai et al., 2021] as our experimented baselines for hypergraph GNNs, as it was a classical and representative work on hypergraph GNNs, and have been used as baselines in many hypergraph GNN works [Chien at el., 2022; Wang et al., 2023]. According to your recommendations, we have now also included the results of ED-HNN [Wang et al., 2023] in our revised paper, which are also reported below:
>
> | MUSAE Dataset | ED-HNN Accuracy | Best Simple GNN Accuracy  | Best Hyper GNN Accuracy | Best LP-GNN Accuracy |
> | --- | --- | --- | --- | --- |
> | GitHub | 0.862 ± 0.001 | 0.872 ± 0.000 (GCN) | 0.808 ± 0.001 (HyperConv) | 0.872 ± 0.000 (GCN+HyperConv) |
> | Facebook | 0.861 ± 0.004 | 0.902 ± 0.002 (GraphSAGE) | 0.792 ± 0.001 (HyperConv) | 0.905 ± 0.000 (GAT+HyperConv) |
> | Twitch_FR | 0.629 ± 0.011 | 0.625 ± 0.001 (GATv2) | 0.624 ± 0.002 (HyperConv) | 0.626 ± 0.000 (GCN+HyperConv) |
> | Twitch_EN | 0.603 ± 0.011 | 0.620 ± 0.003 (GCN) | 0.587 ± 0.007 (HyperConv) | 0.608 ± 0.003 (GAT+HyperConv) |
> | Twitch_ES | 0.722 ± 0.013 | 0.721 ± 0.004 (GCN) | 0.715 ± 0.001 (HyperConv) | 0.729 ± 0.001 (GCN+HyperConv) |
> | Twitch_PT | 0.695 ± 0.015 | 0.689 ± 0.006 (GCN) | 0.701 ± 0.005 (HyperConv) | 0.696 ± 0.004 (GCN+HyperConv) |
> | Twitch_RU | 0.751 ± 0.015 | 0.746 ± 0.002 (GraphSAINT) | 0.743 ± 0.000 (HyperAtten) | 0.744 ± 0.000 (GCN+HyperConv) |
> | Twitch_DE | 0.681 ± 0.006 | 0.657 ± 0.001 (GraphSAGE) | 0.654 ± 0.002 (HyperConv) | 0.657 ± 0.001 (GCN+HyperConv) |
>
> | GRAND Datasets | ED-HNN Accuracy | Best Simple GNN Accuracy | Best Hyper GNN Accuracy | Best LP-GNN Accuracy |
> | --- | --- | --- | --- | --- |
> | ArteryAorta | 0.667 ± 0.013 | 0.630 ± 0.000 (GATv2) | 0.647 ± 0.003 (HyperAtten) | 0.649 ± 0.001 (GAT+HyperConv) |
> | ArteryCoronary | 0.667 ± 0.015 | 0.663 ± 0.000 (GAT) | 0.670 ± 0.003 (HyperAtten) | 0.664 ± 0.001 (GAT+HyperConv) |
> | Breast | 0.661 ± 0.014 | 0.644 ± 0.000 (GraphSAGE) | 0.645 ± 0.001 (HyperConv) | 0.657 ± 0.001 (GAT+HyperConv) |
> | Brain | 0.663 ± 0.013 | 0.625 ± 0.000 (GCN) | 0.632 ± 0.003 (HyperAtten) | 0.645 ± 0.001 (GAT+HyperConv) |
> | Leukemia | 0.603 ± 0.013 | 0.604 ± 0.016 (GraphSAGE) | 0.593 ± 0.008 (HyperAtten) | 0.604 ± 0.004 (GCN+HyperConv) |
> | Lung | 0.670 ± 0.010 | 0.650 ± 0.000 (GCN) | 0.661 ± 0.004 (HyperAtten) | 0.662 ± 0.002 (GAT+HyperConv) |
> | Stomach | 0.661 ± 0.006 | 0.643 ± 0.000 (GCN) | 0.643 ± 0.000 (HyperConv) | 0.656 ± 0.002 (GAT+HyperConv) |
> | LungCancer | 0.602 ± 0.014 | 0.615 ± 0.015 (GraphSAGE) | 0.608 ± 0.008 (HyperAtten) | 0.618 ± 0.003 (GAT+HyperConv) |
> | StomachCancer | 0.610 ± 0.011 | 0.602 ± 0.007 (GCN) | 0.604 ± 0.022 (HyperAtten) | 0.621 ± 0.002 (GAT+HyperConv) |
> | KidneyCancer | 0.603 ± 0.011 | 0.596 ± 0.014 (GraphSAGE) | 0.578 ± 0.012 (HyperAtten) | 0.621 ± 0.002 (GAT+HyperConv) |
>
> | Amazon Dataset | ED-HNN Accuracy | Best Simple GNN Accuracy  | Best Hyper GNN Accuracy | Best LP-GNN Accuracy |
> | --- | --- | --- | --- | --- |
> | Photo | 0.786 ± 0.012 | 0.512 ± 0.141 (GraphSAINT) | 0.465 ± 0.033 (HyperAtten) | 0.715 ± 0.003 (GAT+HyperConv) |
> | Computer | 0.973 ± 0.002 | 0.875 ± 0.020 (GraphSAINT) | 0.842 ± 0.020 (HyperConv) | 0.930 ± 0.007 (GAT+HyperConv) |
>
> The performance of ED-HNN is indeed better on HyperConv and HyperAtten, and becomes the only hypergraph GNN that outperforms simple graph GNNs on the GRAND datasets. In general, the results of ED-HNN is still consistent with the findings in our paper: it outperforms simple graph GNNs on GRAND-Tissues and Amazon, equally on MUSAE-Twitch and GRAND-Diseases, and underperform on MUSAE-GitHub and MUSAE-Facebook. In addition, our LP-GNNs, despite using only HyperConv as its hypergraph GNN component (which is less powerful than ED-HNN), can still outperform or match the performance of ED-HNN in many datasets, which clearly reinforce our claim that combining simple edge and hyperedge information can significantly improve complex graph learning performance. We believe that more appropriate and advanced hypergraph methods, such as ED-HNN, can also be combined with LP-GNN to further improve its performance, but this is beyond the scope of this paper.
>
> (To be continued in the next response)

---

> ### Author Response · Authors · 2023-11-22
> **Response to Reviewer Smxv (3)**
>
> (Continued from the previous response)
>
> ### 5&6. In what ways do the hyperedges complement or enhance the information captured by naturally occurring edges, such as mutual followers in social networks and regulatory effects between genes in biological networks? In addition to Table 3, are there specific examples or case studies where this integration has led to unique insights or improved predictions compared to utilising simple edges alone?
>
> In the MUSAE datasets of SHGB, our hyperedges extracts the clustering information from the simple edges, which serves as a complement to the simple edges, as otherwise algorithms wishing to utilise hyperedge information would have to explicitly compute those hyperedges.
>
> In the GRAND and Amazon datasets of SHGB, the geometric information of genes and image embeddings of the products are not captured by their simple edges or node embeddings, so our hyperedges provide not only a complement to the simple edges, but also an enhancement to the overall information, thereby helps in improving the learning performance.
>
> In addition to Table 3, we have also provided full results of LP-GNNs on all datasets in Tables 12-17 of Appendix C. Notably, LP-GAT+HyperConv and LP-GCN+HyperConv surpass the other methods in 18 of the 23 graphs. These results underscore the benefits of multi-level information integration in hypergraph representation learning. Besides, LP-GAT+HyperConv and LP-GCN+HyperConv also consistently outperform LP-GCN+GAT, showing that it is genuinely the integration of both simple edge and hyperedge information that contributes to the improved prediction performance, rather than arbitrarily integrating two node embeddings.
>
> We have now included the clarifications of whether these hyperedges are complementary or enhancing in Section 3.2 of our paper.
>
> We hope our response and additional results can sufficiently resolve your concerns. We sincerely appreciate it if you could kindly consider improving the score, and are very happy to answer any further questions you may have.
>
> [Bai et al., 2021]: Hypergraph convolution and hypergraph attention, Pattern Recognition'21
>
> [Chien et al., 2022]: You are AllSet: A Multiset Function Framework for Hypergraph Neural Networks, ICLR'22
>
> [Wang et al., 2023]: Equivariant Hypergraph Diffusion Neural Operators, ICLR'23

---

### Official Review · Reviewer_3S38 · 2023-10-31

**Soundness:** 2 fair
**Presentation:** 3 good
**Contribution:** 2 fair
**Rating:** 5
**Confidence:** 3

**Summary:**

The paper proposes datasets that contains hyperedges and simple edges for GNNs. The paper reads well but motivation for proposing the
datasets combining hypergraphs and simple edges is unclear.  Datasets are not from the real-world. Insights for future research are not unclear.  Authors should fix the download links for codes and datasets.

**Strengths:**

S1. Datasets cover various domains.

S2. The paper reads well.

**Weaknesses:**

W1. Motivation for proposing semi-hypergraphs is unclear.  Real-world applications for semi-hypergraphs are not discussed. Or importance of semi-hypergraphs for future research is not discussed. See Q1.

W2. Datasets are not effective for GNNs. See Q2.

W3. Datasets are not from the real-world but generated by rules and algorithms. See Q3.

W4. Insights for future research is not clear. See Q4.

**Questions:**

Q1: Can authors show real-world applications for semi-hypergraphs? Or can authors show evidences for that proposing semi-hypergraphs is important for future research?

Q2: Figure 4 (a) shows that for most of the datasets, Hyper GNNs and simple GNNs have almost the same accuracy. Can authors give comments on this?

Q3: Can authors explain why not search for real-word datasets but generate by rules or algorithms?

Q4: Can authors show the insights for future research?

---

> ### Author Response · Authors · 2023-11-22
> **Response to Reviewer 3S38 (1)**
>
> Thank you for the attentive review and insightful questions. Please kindly see below for our responses to your comments:
>
> ### Q1: Can authors show real-world applications for semi-hypergraphs? Or can authors show evidences for that proposing semi-hypergraphs is important for future research?
>
> Most real-world complex networks preserve a mixture of pairwise node relations (simple edges) and local clustering node relations (hyperegdes). The two different relations not only capture different information about the graph, but also normally come from different modalities. This makes the semi-hypergraphs a valuable augmentation to simple graphs by allowing them to contain hyperedges, in order to capture information in both edge types. In addition to the semi-hypergraphs provided by SHGB:
>
> - Social networks that contain both mutual connections (simple edges) and friend circles (hyperedges);
> - Gene regulatory networks that contain both regulatory effects (simple edges) and geometric information between nearby genes (hyperedges);
> - E-commerce networks that contain both co-purchase and co-review relations (simple edges) and similarity between the images of the products (hyperedges),
>
> There are also many real-world applications for semi-hypergraphs including:
>
> - Citation networks with citations between two papers (simple edges) and multiple papers containing the same author (hyperegdes) [1];
> - Biological networks with pairwise associations and interactions between chemicals, diseases, genes and phenotypes (simple edges), are research articles where multiple nodes (i.e. chemicals, diseases, genes, phenotypes) and edges (i.e., pairwise associations and interactions) are studied (hyperedges) [2];
> - Product networks where simple edges connects the same products of different brands or the different products manufactured by the same brand, and hyperedges contain sets of products bought on the same shopping trips [1].
>
> Since the datasets in our SHGB already cover the same or similar domains of the above-mentioned applications, we did not include them in the first version of SHGB. However, we shall endeavour to update SHGB on a regular basis to include more real-world semi-hypergraphs in diverse domains.
>
> ### Q2: Figure 4(a) shows that for most of the datasets, hyper GNNs and simple GNNs have almost the same accuracy. Can authors give comments on this?
>
> The scatter plot in Figure 4(a) illustrates the comparisons of accuracies between the *best-performing* hypergraph GNNs and simple graph GNNs, rather than all GNNs. It shows that the best-performing hypergraph GNNs match the best-performing simple graph GNNs’ accuracies on GRAND, underperform on MUSAE, and outperform on Amazon. We would like to kindly direct the reviewer’s attention to Tables 6–11 in Appendix C for the detailed results of all GNNs on SHGB, which clearly demonstrate distinct performance levels among the GNNs, thereby indicating that the SHGB datasets are indeed effective for GNNs.
>
> We have now updated the caption of Figure 4 to enhance clarity and avoid confusion.
>
> ### Q3: Can authors explain why not search for real-word datasets but generate by rules or algorithms?
>
> Existing hypergraph datasets can be divided into two main categories in terms of how hyperegdes are constructed: ground-truth based and rule based. While ground-truth based dataset do exist, they are usually limited in specific domains, and normally do not support the hyperedges to be constructed from a different data modality than simple edges. The rule based hypergraph datasets also serve as an important part of the benchmarks and cannot be neglected [4].
>
> Our datasets, while closer to the second category, provide more concrete rules that can be regarded as “almost” ground-truths, rather than just looking at the embedding space (MUSAE & GRAND), and allow the simple edges and hyperedges to be formed from different modalities (Amazon). We shall endeavour to update SHGB on a regular basis to include more real-world semi-hypergraphs with hyperegdes constructed under ground truths.
>
> We have now included the above explanations in Sections 2 and 3.2 of our paper to enhance its clarity.
>
> [1] Ilya Amburg, Nate Veldt, and Austin R. Benson. Clustering in graphs and hypergraphs with categorical edge labels. WWW 2020.
>
> [2] Davis AP, Wiegers TC, Johnson RJ, Sciaky D, Wiegers J, and Mattingly CJ. Comparative Toxicogenomics Database (CTD): update 2023. Nucleic Acids Res. URL: http://ctdbase.org/
>
> [3] Philip S. Chodrow, Nate Veldt, and Austin R. Benson. Generative hypergraph clustering: from blockmodels to modularity. Science Adwvances, 2021.
>
> [4] Yifan Feng, Haoxuan You, Zizhao Zhang, Rongrong Ji, Yue Gao. Hypergraph neural networks. AAAI 2019.
>
> (To be continued in the next response)

---

> ### Author Response · Authors · 2023-11-22
> **Response to Reviewer 3S38 (2)**
>
> (Continued from the previous response)
>
> ### Q4: Can authors show the insights for future research?
>
> We would like to highlight insights for future research from the following two perspectives:
>
> 1. Datasets and benchmarks:
>     - We bring the notion of unifying the benchmarks for GNNs on different types of graphs (simple graphs, hypergraphs, and potentially other higher-order graphs), so that the graph representation learning community can use SHGB to conveniently and effectively evaluate whether a particular model designed to capture hyperedge information can indeed leverage the graph learning performance;
>     - To facilitate convenient and flexible evaluation under SHGB, we have built a concrete Python library for the users to access our datasets and evaluate within our framework, which is installable via pip (we are unable to disclose the library name for anonymity). We have mentioned this in Appendix D of our paper.
> 2.  Algorithms:
>     - With the simple model (LP-GNNs) integrating simple edge and hyperedge information that can already improve the node prediction performance, we would like to encourage the community to have more in-depth investigations on improving hypergraph learning performance by combining simple edge and hyperedge information;
>     - We build the Amazon datasets where simple edges and hyperedges are constructing from different data modalities, in order to show the research insights of multimodal learning on graphs, and combining information from different modalities for improving graph learning performance.
>
> We have now included the above insights in the Conclusion section of our paper, in order to make those insights explicit, and to enhance the clarity of our paper.
>
> We sincerely appreciate it if you could kindly consider improving the scores if the above response can sufficiently address your concerns. We are very happy to answer any further questions you may have.

---

### Official Review · Reviewer_RFzy · 2023-11-01

**Soundness:** 2 fair
**Presentation:** 3 good
**Contribution:** 2 fair
**Rating:** 5
**Confidence:** 4

**Summary:**

This work aims to provide a more complete evaluation of hypergraph deep learning models. The approach includes a) constructing new hypergraph datasets that consist of both simple edges and clique-based hyperedges, b) comparing hypergraph neural networks and graph neural networks, and c) properly combining the two types of neural networks. The work also studies hypergraph sampling approaches for hypergraph NNs.

**Strengths:**

+ It is a valid idea to build hypergraph datasets that hold both simple edges and hyperedges.

+ It is novel to investigate hypergraph sampling approaches for hypergraph neural networks.

+ It is reasonable to combine GNNs and hypergraph NNs to achieve overall best performance.

**Weaknesses:**

- The work misses some solid foundations. First, the work seems unaware of how practical hypergraphs are typically models. The work only discusses those datasets used to evaluate hypergraph NNs but really misses the discussion on the entire area that studies hypergraph modeling, higher-order graphs for data analysis, e.g. [1][2].

- Because the work is unaware of that area. The claim that "using hyperedges may overlook simple pairwise node relations and thus make hypergraphs substantially lose useful graph information" is overclaimed. Properly modeling hyperedges as sets and using complex set functions essentially cover simple pairwise relations as a special case [3]. The set representation is much more powerful in principle. The not-idea performance of hypergraph NNs as opposed to GNNs is just due to the non-idea way to construct hypergraphs and the suboptimality of hypergraph NNs.

- Some recent more principled hypergraph NNs are missing to discuss, e.g. [4][5]. This is a weak point for a benchmark paper.

[1] Higher-order organization of complex networks, Science 2016

[2] Networks beyond pairwise interactions: Structure and dynamics, Physics Report, 2020

[3] Submodular hypergraphs: p-laplacians, cheeger inequalities and spectral clustering, ICML 2018

[4] Unignn: a unified framework for graph and hypergraph neural networks, IJCAI 2021

[5] Equivariant Hypergraph Diffusion Neural Operators, ICLR 2023

**Questions:**

no specific questions.

The authors are suggested to extensively discuss relevant works to address the listed weaknesses.

---

> ### Author Response · Authors · 2023-11-22
> **Response to Reviewer RFzy (1)**
>
> Thank you for the thoughtful questions. Please kindly see below for our responses to your comments:
>
> ### The work misses some solid foundations. First, the work seems unaware of how practical hypergraphs are typically models. The work only discusses those datasets used to evaluate hypergraph NNs but really misses the discussion on the entire area that studies hypergraph modeling, higher-order graphs for data analysis, e.g. [1][2]. Because the work is unaware of that area. The claim that "using hyperedges may overlook simple pairwise node relations and thus make hypergraphs substantially lose useful graph information" is overclaimed. Properly modeling hyperedges as sets and using complex set functions essentially cover simple pairwise relations as a special case [3]. The set representation is much more powerful in principle. The not-idea performance of hypergraph NNs as opposed to GNNs is just due to the non-idea way to construct hypergraphs and the suboptimality of hypergraph NNs.
>
> Our work is focused on the practical, data-driven ML applications on complex networks, and unifying the benchmarks of GNNs across different graph types. Based on this core focus of this work, we argue that simple edges should not be only taken as a special case, but rather a parallel input and let the learning mechanics to determine which information is worth extracting. Allowing graphs to contain simple edges have the following benefits in ML applications:
>
> - As discussed in the Introduction section of our paper, although hypergraphs can include simple edges by using hyperedges with only two nodes, this is not a principled approach of utilising hyperedges, especially when they are employed to described clustering or homogeneity relations between multiple nodes from another perspective than the simple edges, and can still potentially hurt ML systems’ task performance.
> - Existing works on hypergraph GNNs and simple graph GNNs are normally evaluated separately on different datasets, and it is highly desired to develop a unified benchmark to evaluate whether hypergraph GNNs can indeed enhance learning performance on complex graphs than simple GNNs. This would also enable works on integrating simple edge and hyperedge information to be reliably trained and evaluated.
> - By allowing graphs to contain both hyperedges and simple edges, we can also enable the hyperedges and simple edges to be constructed from different perspectives (as in our GRAND datasets, where simple edges are constructed from pairwise gene regulatory effects, and hyperedges represent geometric information of nearby genes) or even different data modalities (as in our Amazon datasets, where hyperegdes are constructed from the image modality, and the node embeddings are collected from the text modality). This would allow the graph to capture much more concrete information about the real-world network, and enable works on those datasets to enhance their learning performance. With this notion, we also bring insights to the community of combining different types of edge information from different modalities, and perform multimodal learning on complex networks.
>
> [1] Higher-order organization of complex networks, Science 2016
>
> [2] Networks beyond pairwise interactions: Structure and dynamics, Physics Report, 2020
>
> [3] Submodular hypergraphs: p-laplacians, cheeger inequalities and spectral clustering, ICML 2018
>
> (To be continued in the next response)

---

> ### Author Response · Authors · 2023-11-22
> **Response to Reviewer RFzy (2)**
>
> (Continued from the previous response)
>
> ### Some recent more principled hypergraph NNs are missing to discuss, e.g. [4][5]. This is a weak point for a benchmark paper.
>
> Thank you very much for pointing out the more recent hypergraph GNNs in addition to the models that we have originally included [6]. We originally selected [6] as our experimented baselines for hypergraph GNNs, as it was a classical and representative work on hypergraph GNNs, and have been used as baselines in many hypergraph GNN works [5][7]. According to your recommendations, we have now also included the accuracies of ED-HNN [5] in our revised paper, which are also reported below:
>
> | MUSAE Dataset | ED-HNN Accuracy | Best Simple Graph GNN Accuracy  | Best Hypergraph GNN Accuracy | Best LP-GNN Accuracy |
> | --- | --- | --- | --- | --- |
> | GitHub | 0.862 ± 0.001 | 0.872 ± 0.000 (GCN) | 0.808 ± 0.001 (HyperConv) | 0.872 ± 0.000 (LP-GCN+HyperConv) |
> | Facebook | 0.861 ± 0.004 | 0.902 ± 0.002 (GraphSAGE) | 0.792 ± 0.001 (HyperConv) | 0.905 ± 0.000 (LP-GAT+HyperConv) |
> | Twitch_FR | 0.629 ± 0.011 | 0.625 ± 0.001 (GATv2) | 0.624 ± 0.002 (HyperConv) | 0.626 ± 0.000 (LP-GCN+HyperConv) |
> | Twitch_EN | 0.603 ± 0.011 | 0.620 ± 0.003 (GCN) | 0.587 ± 0.007 (HyperConv) | 0.608 ± 0.003 (LP-GAT+HyperConv) |
> | Twitch_ES | 0.722 ± 0.013 | 0.721 ± 0.004 (GCN) | 0.715 ± 0.001 (HyperConv) | 0.729 ± 0.001 (LP-GCN+HyperConv) |
> | Twitch_PT | 0.695 ± 0.015 | 0.689 ± 0.006 (GCN) | 0.701 ± 0.005 (HyperConv) | 0.696 ± 0.004 (LP-GCN+HyperConv) |
> | Twitch_RU | 0.751 ± 0.015 | 0.746 ± 0.002 (GraphSAINT) | 0.743 ± 0.000 (HyperAtten) | 0.744 ± 0.000 (LP-GCN+HyperConv) |
> | Twitch_DE | 0.681 ± 0.006 | 0.657 ± 0.001 (GraphSAGE) | 0.654 ± 0.002 (HyperConv) | 0.657 ± 0.001 (LP-GCN+HyperConv) |
>
> | GRAND Datasets | ED-HNN Accuracy | Best Simple Graph GNN Accuracy | Best Hypergraph GNN Accuracy | Best LP-GNN Accuracy |
> | --- | --- | --- | --- | --- |
> | ArteryAorta | 0.667 ± 0.013 | 0.630 ± 0.000 (GATv2) | 0.647 ± 0.003 (HyperAtten) | 0.649 ± 0.001 (LP-GAT+HyperConv) |
> | ArteryCoronary | 0.667 ± 0.015 | 0.663 ± 0.000 (GAT) | 0.670 ± 0.003 (HyperAtten) | 0.664 ± 0.001 (LP-GAT+HyperConv) |
> | Breast | 0.661 ± 0.014 | 0.644 ± 0.000 (GraphSAGE) | 0.645 ± 0.001 (HyperConv) | 0.657 ± 0.001 (LP-GAT+HyperConv) |
> | Brain | 0.663 ± 0.013 | 0.625 ± 0.000 (GCN) | 0.632 ± 0.003 (HyperAtten) | 0.645 ± 0.001 (LP-GAT+HyperConv) |
> | Leukemia | 0.603 ± 0.013 | 0.604 ± 0.016 (GraphSAGE) | 0.593 ± 0.008 (HyperAtten) | 0.604 ± 0.004 (LP-GCN+HyperConv) |
> | Lung | 0.670 ± 0.010 | 0.650 ± 0.000 (GCN) | 0.661 ± 0.004 (HyperAtten) | 0.662 ± 0.002 (LP-GAT+HyperConv) |
> | Stomach | 0.661 ± 0.006 | 0.643 ± 0.000 (GCN) | 0.643 ± 0.000 (HyperConv) | 0.656 ± 0.002 (LP-GAT+HyperConv) |
> | Lungcancer | 0.602 ± 0.014 | 0.615 ± 0.015 (GraphSAGE) | 0.608 ± 0.008 (HyperAtten) | 0.618 ± 0.003 (LP-GAT+HyperConv) |
> | Stomachcancer | 0.610 ± 0.011 | 0.602 ± 0.007 (GCN) | 0.604 ± 0.022 (HyperAtten) | 0.621 ± 0.002 (LP-GAT+HyperConv) |
> | KidneyCancer | 0.603 ± 0.011 | 0.596 ± 0.014 (GraphSAGE) | 0.578 ± 0.012 (HyperAtten) | 0.621 ± 0.002 (LP-GAT+HyperConv) |
>
> | Amazon Dataset | ED-HNN Accuracy | Best Simple Graph GNN Accuracy  | Best Hypergraph GNN Accuracy | Best LP-GNN Accuracy |
> | --- | --- | --- | --- | --- |
> | Photo | 0.786 ± 0.012 | 0.512 ± 0.141 (GraphSAINT) | 0.465 ± 0.033 (HyperAtten) | 0.715 ± 0.003 (LP-GAT+HyperConv) |
> | Computer | 0.973 ± 0.002 | 0.875 ± 0.020 (GraphSAINT) | 0.842 ± 0.020 (HyperConv) | 0.930 ± 0.007 (LP-GAT+HyperConv) |
>
> The performance of ED-HNN is indeed better on HyperConv and HyperAtten, and becomes the only hypergraph GNN that outperforms simple graph GNNs on the GRAND datasets. In general, the results of ED-HNN is still consistent with the findings in our paper: it outperforms simple graph GNNs on GRAND-Tissues and Amazon, equally on MUSAE-Twitch and GRAND-Diseases, and underperform on MUSAE-GitHub and MUSAE-Facebook. In addition, our LP-GNNs, despite using only HyperConv as its hypergraph GNN component (which is less powerful than ED-HNN), can still outperform or match the performance of ED-HNN in many datasets, which clearly reinforce our claim that combining simple edge and hyperedge information can significantly improve complex graph learning performance. We believe that more appropriate and advanced hypergraph methods, such as ED-HNN, can also be combined with LP-GNN to further improve its performance, but this is beyond the scope of this paper.
>
> We sincerely appreciate it if you could kindly consider improving the scores if we have sufficiently addressed the concerns. We are very happy to answer any further questions you may have.
>
> [4] Unignn: a unified framework for graph and hypergraph neural networks, IJCAI 2021
>
> [5] Equivariant Hypergraph Diffusion Neural Operators, ICLR 2023
>
> [6] Hypergraph convolution and hypergraph attention, Pattern Recognition 2021
>
> [7] You are AllSet: A Multiset Function Framework for Hypergraph Neural Networks, ICLR 2022

---

### Author Response · Authors · 2023-11-22
**Summary of Rebuttals and Revisions**

We thank all the reviewers for their valuable feedback and insightful suggestions. We would like to clarify and emphasise the following two points:

- The core intuition of bringing the notion of “semi”-hypergraphs is *not* to introduce a new graph type, but is rather an augmentation of the current graph modelling, with the inclusion of hyperedges (or equivalently, an augmentation of hypergraphs with simple edges), so that these two types of edge connections are maintained concurrently. With SHGB, we provide unified benchmarks for GNNs on different types of graphs (simple graphs, hypergraphs, and potentially other higher-order graphs). We also argue that allowing graphs to have both hyperedges and simple edges can enable an improvement in ML performance on complex networks, which are originally modelled as either pure simple graphs or pure hypergraphs.
- By allowing graphs to contain both hyperedges and simple edges, we can also enable the hyperedges and simple edges to be constructed from different perspectives (as in our GRAND datasets, where simple edges are constructed from pairwise gene regulatory effects, and hyperedges represent geometric information of nearby genes) or even different data modalities (as in our Amazon datasets, where hyperegdes are constructed from the image modality, and the node embeddings are collected from the text modality). This flexible hyperedge construction encourages the design of novel graph learning algorithms (such as LP-GNNs) that integrate simple edge and hyperedge information from different perspectives or modalities, and perform multimodal learning on complex networks.

Based on the reviews, we have made the following revisions to our paper:

- We have now included additional results on a recent hypergraph model, ED-HNN [1] (Sections 2, 4.1, 4.2, Appendix C);
- We have added a description of two categories of hypergraph datasets based on how hyperedges are constructed — ground-truth based and rule based (Sections 2, 3.2);
- We have added descriptions of whether the constructed hyperedges are complements of simple edges, or enhancements of the overall hypergraph information, in each group (MUSAE, GRAND, Amazon) of the SHGB datasets (Section 3.2);
- We have now made it explicit about the insights that SHGB can bring to future research directions for the community, from both the datasets & benchmarks and the ML algorithms perspectives (Section 5);
- We have also refined the wording in various places of our paper, to enhance clarity and avoid confusion.

We have also addressed each reviewer’s comments with more detailed, in-depth responses. We sincerely appreciate it if the reviewers could kindly consider improving the scores if we have sufficiently addressed the concerns. Once again, we appreciate all the suggestions made by reviewers to improve our work. It is our pleasure to hear your feedback, and we look forward to answering your follow-up questions.